# Evidence for allocentric boundary and goal direction information in the human entorhinal cortex and subiculum

J.P. Shine [1,4], J.P. Valdés-Herrera[1,4], C. Tempelmann[2] & T. Wolbers[1,2,3]

In rodents, cells in the medial entorhinal cortex (EC) and subiculum code for the allocentric direction to environment boundaries, which is an important prerequisite for accurate positional coding. Although in humans boundary-related signals have been reported, there is no evidence that they contain allocentric direction information. Furthermore, it has not been possible to separate boundary versus goal direction signals in the EC/subiculum. Here, to address these questions, we had participants learn a virtual environment containing four unique boundaries. Participants then underwent fMRI scanning where they made judgements about the allocentric direction of a cue object. Using multivariate decoding, we found information regarding allocentric boundary direction in posterior EC and subiculum, whereas allocentric goal direction was decodable from anterior EC and subiculum. These data provide the first evidence of allocentric boundary coding in humans, and are consistent with recent conceptualisations of a division of labour within the EC.

---

[1] German Center for Neurodegenerative Diseases (DZNE), Aging and Cognition Research Group, 39120 Magdeburg, Germany. [2] Department of Neurology, Medical Faculty, Otto-von-Guericke-University Magdeburg, Magdeburg 39120, Germany. [3] Center for Behavioral Brain Sciences, 39118 Magdeburg, Germany. [4]These authors contributed equally: J.P. Shine, J.P. Valdés-Herrera. Correspondence and requests for materials should be addressed to J.P.S. (email: jonathan.shine@dzne.de)

The entorhinal cortex (EC) provides the primary cortical input to the hippocampus[1]. Given its distinct profile of anatomical connectivity, different subregions of the EC are hypothesised to convey different types of information that is combined in service of episodic memory and spatial navigation[2,3]. In rodents, the medial EC (MEC) receives projections from regions processing spatial information, including the postrhinal cortex (primate parahippocampal cortex; PHC) and subiculum, whereas the rodent lateral EC (LEC) receives information from the object-sensitive perirhinal cortex. This anatomical connectivity is preserved in humans, with posterior EC (homologous with rodent MEC) showing preferential connectivity with PHC[4] and posterior subiculum[5], whereas anterior EC (homologous with rodent LEC) shares preferential connectivity with the perirhinal cortex[4] and anterior subiculum[5]. These posterior and anterior divisions of EC and subiculum, therefore, were hypothesised to code for "where" and "what" information, respectively[6,7]. Consistent with a division of labour in the EC, recent evidence of spatial coding in the rodent LEC led to the proposal that MEC supports neural populations involved in spatial navigation (e.g., grid cells), whereas the LEC codes for external sensory inputs (e.g., landmarks or prominent objects in the environment)[8]. Alternatively, the MEC and LEC may support allocentric and egocentric reference frames, respectively[9].

The rodent MEC contains spatially tuned neural populations, including border cells[10] that fire near environment boundaries, a subset of which indicate the boundary's allocentric direction. The coding of environment boundaries is essential for neural computations determining one's spatial location. For example, path integration (i.e., the ability to update one's spatial position using self-motion cues) invariably accumulates error[11], and boundaries help correct these noisy positional estimates[12] as well as providing strong cues for reorientation[13]. Environment boundaries are vital for other spatially tuned neurons in the MEC. Specifically, grid-cell firing fields anchor to an enclosure's walls[14], with irregular-shaped enclosures deforming the grid cell's characteristic hexagonal symmetry[15]. In the rodent hippocampus, the removal of environment boundaries leads to the degradation of place cells[16], while expanding the size of an environment by moving its boundaries results in commensurate expansion of place cell firing fields[17]. Cells coding for environment boundaries were identified also in the rodent dorsal subiculum (homologous with human posterior subiculum), with these so-called boundary vector cells containing information about both the allocentric direction and distance[18] to walls. Accordingly, place cell activity has been modelled as the summed and thresholded input of these spatial properties describing the boundary position[19]. In sum, boundary coding is a fundamental component of spatial navigation.

In humans, boundaries are behaviourally salient, aiding reorientation[13] and used to define object locations[20,21]. Recordings from the EC in intracranial patients revealed stronger grid-cell-like representations[22] when participants navigated near to walls in a virtual environment (VE). Similarly, boundaries engage the hippocampus[23], with univariate activity increasing with the number of boundary elements to-be-imagined[24]. Intracranial recordings implicate the subiculum as the locus of this boundary coding[25], with increased theta frequency activity associated with object locations proximal to the VE's walls. Despite evidence of boundary-related signals in the EC/hippocampus, these studies do not demonstrate the allocentric directional information necessary to support accurate positional coding[19]. Specifically, it is unclear whether the signal discriminates if the boundary is located to the North, South, East or West, regardless of the participant's position and orientation in the environment. Indirect evidence of allocentric boundary coding in the EC/subiculum was shown in a recent functional magnetic resonance imaging (fMRI) study using multivariate analysis methods, in which an allocentric goal direction signal, thought to reflect the simulated representation of head direction, was identified[26]. The heading direction to-be-imagined, however, was determined by goal objects arranged in front of boundaries in the enclosed VE. Consequently, it was not possible to disentangle whether this signal reflected allocentric goal or allocentric boundary direction. Moreover, this study lacked the anatomical resolution to differentiate subregions of the EC and subiculum. An outstanding question, therefore, is whether there is an allocentric boundary direction signal in the human EC and subiculum, and if this is distinct from allocentric goal direction coding.

In contrast to the MEC, cells in rodent LEC show far less spatial tuning. LEC neurons code space in the presence of objects, for example, showing trace memory for object locations[27]. The function of the LEC, therefore, may be to code for local environmental information, such as the positions of objects, constituting a landmark, or goal location[8]. Moreover, in contrast to MEC, spatially tuned neurons in the LEC seem to respond in an egocentric reference frame[9]. In humans, a preference for object versus scene/spatial manipulations has been demonstrated in anterior EC[6,7], reflecting its strong connections with object-specific perirhinal cortex. As noted above, previous studies have lacked the anatomical resolution to determine where in the EC/subiculum allocentric goal direction representations are located[26]. One possibility is that the human anterior EC supports spatial judgements regarding the allocentric direction of landmarks in the environment (i.e., object-in-place coding).

In the current study, we investigated the neural correlates of allocentric boundary and goal direction coding in the human medial temporal lobe, using immersive virtual reality and high-resolution fMRI. Importantly, we used anatomically informed masks of the EC and subiculum, and orthogonalised allocentric boundary and goal processing. Unlike previous studies demonstrating increased neural activity associated with boundaries, we used multivariate analyses to determine whether different allocentric boundary directions (i.e., North, South, East and West) are associated with different evoked responses, facilitating classification of these directions. Given the proposed division of labour within the EC and subiculum, we separated our ROIs into anterior and posterior portions to examine longitudinal differences in decoding accuracy according to task. Specifically, we tested whether posterior EC and subiculum contributed to allocentric boundary coding. Furthermore, we investigated whether anterior EC supports representations of allocentric goal direction given its potential involvement in the coding of landmark objects.

In line with the proposed division of labour in EC/subiculum, we show that the posterior EC and subiculum code for allocentric boundary direction whereas anterior regions code for the allocentric direction towards a goal. These data provide evidence for a fundamental component of place coding.

## Results

**Behavioural.** Before proceeding to the scanned fMRI task, participants were required to learn the layout of the VE as assessed by a judgement of relative direction (JRD) task. Participants needed on average 3.39 (standard deviation = 1.01) rounds of exploration to learn the relative directions of the global landmarks. To test whether performance on the JRD changed as a function of the initial landmark facing direction, or the angular disparity of the second landmark relative to the first, accuracy and RT data were submitted to separate repeated-measures ANOVAs comprising the factors Landmark (Mountain, Cathedral, Clock tower and City) and Angular Disparity (90°, 180° and 270°). Accuracy was modulated by the angular disparity of the second landmark [repeated-measures

ANOVA: $F$ (2, 54) = 9.49, $p$ = 0.0003, $\eta_p^2$ = 0.26], but not the initial landmark facing direction [repeated-measures ANOVA: $F$ (3, 81) = 0.41, $p$ = 0.74, $\eta_p^2$ = 0.015] and these two factors did not interact [repeated-measures ANOVA: $F$ (6, 162) = 0.34, $p$ = 0.92, $\eta_p^2$ = 0.012] (Fig. 1a). Performance was significantly better for landmarks located at 180°, or 90° disparity versus those located at 270° disparity ([paired-sample $t$ test: $t$ (27) = 3.45, $p$ = 0.001, 95% CI: [0.05–0.18], Hedges's $g_{av}$ = 0.84; adjusted alpha = 0.05/3 comparisons = 0.017] and [paired-sample $t$ test: $t$ (27) = 3.66, $p$ = 0.001, 95% CI: [0.04–0.14], Hedges's $g_{av}$ = 0.73], respectively); performance did not differ between 180° and 90° disparities [paired-sample $t$ test: $t$ (27) = 0.93, $p$ = 0.36 95% CI: [−0.07 to 0.03], Hedges's $g_{av}$ = 0.18]. The same analysis for RT data revealed a similar pattern of results, with a main effect of angular disparity [repeated-measures ANOVA: $F$ (1.16–31.36) = 9.37, $p$ = 0.003, $\eta_p^2$ = 0.257], but no effect of initial landmark facing direction [repeated-measures ANOVA: $F$ (1.44, 38.98) = 0.73, $p$ = 0.45, $\eta_p^2$ = 0.026] and no interaction between these factors [repeated-measures ANOVA: $F$ (2.3, 62.21) = 0.61, $p$ = 0.57, $\eta_p^2$ = 0.022]. Responses were fastest for landmarks located at 180° disparity versus those located at 270 [paired-sample $t$ test: $t$ (27) = 3.27, $p$ = 0.003, 95% CI: [−4.29 to −0.98], Hedges's $g_{av}$ = 0.70; adjusted alpha = 0.05/3 comparisons = 0.017] with a trend towards being faster than those located at 90° [paired-sample $t$ test: $t$ (27) = 2.4, $p$ = 0.02, 95% CI: [−2.7 to −0.21], Hedges's $g_{av}$ = 0.48]; responses for landmarks located at 90° disparity were faster than those located at 270° [paired-sample $t$ test: $t$ (27) = 3.71, $p$ = 0.0009, 95% CI: [−1.83 to −0.53], Hedges's $g_{av}$ = 0.26].

Participants were required to understand that the VE boundaries were impassable. It was crucial, therefore, that they spent a large proportion of their exploration time in close proximity to the boundaries. To assess time spent near to the boundaries, for each participant we divided the total explorable space of the VE into 10,000 bins (each bin = 5 × 5 virtual metres$^2$ of the VE), and normalised the total time across all rounds of exploration so that it summed to 1. We then created masks of the areas next to the boundaries in the VE, comprising the four different allocentric boundary directions (each mask per boundary side was 40 × 5 virtual metres). On average, participants spent 76.16% (SD = 10.45%) near the environment boundaries (Fig. 1b). A one-way repeated-measures ANOVA revealed that exploration time did not differ as a function of allocentric boundary direction [repeated-measures ANOVA: $F$ (2.27, 61.24) = 1.16, $p$ = 0.33, $\eta_p^2$ = 0.041].

Given the extensive training required prior to fMRI scanning, performance on the scanner task was very high (Fig. 1c). Accuracy and RT data were submitted to separate one-way repeated-measures ANOVAs comprising allocentric goal direction (North, South, East and West). Accuracy was matched across all four allocentric goal directions [repeated-measures ANOVA: $F$ (2.18, 58.89) = 0.35, $p$ = 0.73, $\eta_p^2$ = 0.013], whereas RT differed [repeated-measures ANOVA: $F$ (3, 81) = 29.18, $p$ = 0.00001, $\eta_p^2$ = 0.52]. Follow-up comparisons revealed that responses to allocentric goals located to the North were significantly quicker than those goals located to the South [paired-sample $t$ test: $t$ (27) = 3.15, $p$ = 0.004, 95% CI: [−0.06 to −0.01], Hedges's $g_{av}$ = 0.36; adjusted alpha = 0.05/6 comparisons = 0.008], East [paired-sample $t$ test: $t$ (27) = 6.72, $p$ < 0.001, 95% CI: [−0.09 to −0.05], Hedges's $g_{av}$ = 0.65] and West [paired-sample $t$ test: $t$ (27) = 10.20, $p$ < 0.001, 95% CI: [−0.12 to −0.08], Hedges's $g_{av}$ = 0.85]. Similarly, responses to goals located to the South were faster than those located to the East [paired-sample $t$ test: $t$ (27) = 3.12, $p$ = 0.004, 95% CI: [−0.05 to −0.01], Hedges's $g_{av}$ = 0.3] and West [paired-sample $t$ test: $t$ (27) = 5.06, $p$ = 0.00003, 95% CI: [−0.09 to −0.04], Hedges's $g_{av}$ = 0.54]; differences in RTs for responses to goals located to the East and West did not survive Bonferroni-

correction [paired-sample $t$ test: $t$ (27) = 2.45, $p$ = 0.021, 95% CI: [−0.06 to −0.01], Hedges's $g_{av}$ = 0.26]. These differences in RT may reflect participants forming a reference frame in the environment, with the Mountain and Cathedral providing a conceptual North–South axis. Consequently, responses to allocentric goal judgements in these directions may be facilitated[28,29]. Importantly, however, these differences in RT did not influence subsequent decoding performance (see Supplementary Note 1). Although the participants performed a task regarding the allocentric goal direction, trials could be coded also according to the allocentric boundary direction. The same repeated-measures ANOVAs were conducted with this coding, and again revealed that accuracy was matched across allocentric boundary direction [repeated-measures ANOVA: $F$ (3, 81) = 1.87, $p$ = 0.14, $\eta_p^2$ = 0.065], but that RT differed [repeated-measures ANOVA: $F$ (3, 81) = 8.24, $p$ = 0.00007, $\eta_p^2$ = 0.234]. The differences in RT stemmed from faster responses to trials in which the allocentric boundary was located to the North relative to East [paired-sample $t$ test: $t$ (27) = 3.34, $p$ = 0.002, 95% CI: [−0.04 to −0.01], Hedges's $g_{av}$ = 0.23; adjusted alpha = 0.05/6 comparisons = 0.008] and West [paired-sample $t$ test: $t$ (27) = 2.91, $p$ = 0.007, 95% CI: [−0.04 to −0.01], Hedges's $g_{av}$ = 0.19]. The same pattern of data were evident for allocentric boundaries located to the South, with faster responses relative to East [paired-sample $t$ test: $t$ (27) = 4.08, $p$ = 0.0003, 95% CI: [−0.05 to −0.02], Hedges's $g_{av}$ = 0.28] and West [paired-sample $t$ test: $t$ (27) = 3.45, $p$ = 0.002, 95% CI: [−0.04 to −0.01], Hedges's $g_{av}$ = 0.24]; RTs did not differ for allocentric boundaries located to the North and South [paired-sample $t$ test: $t$ (27) = 0.75, $p$ = 0.46, 95% CI: [−0.01 to 0.02], Hedges's $g_{av}$ = 0.06], nor East and West [Paired-sample $t$ test: $t$ (27) = 0.69, $p$ = 0.49, 95% CI: [−0.01 to 0.01], Hedges's $g_{av}$ = 0.04] (Supplementary Fig. 1).

**Multivariate decoding.** Mean group-level decoding accuracy of allocentric boundary direction in the posterior EC was significantly above chance (Fig. 2a), as determined by the bias-corrected and accelerated boot-strap (BCa) null distribution [non-parametric Monte Carlo significance test: $p$ = 0.0005]. The same analysis for allocentric goal direction, however, did not exceed chance performance [non-parametric Monte Carlo significance test: $p$ = 0.5153]. In the anterior EC (Fig. 2b), it was possible to decode also allocentric boundary direction [non-parametric Monte Carlo significance test: $p$ = 0.0239], however, in contrast to the posterior EC, the anterior EC contained information also regarding the allocentric goal direction [non-parametric Monte Carlo significance test: $p$ = 0.0022].

The pattern of data in the posterior subiculum mirrored that of the posterior EC, with significantly above chance decoding of allocentric boundary direction [non-parametric Monte Carlo significance test: $p$ = 0.0016], but not allocentric goal direction [non-parametric Monte Carlo significance test: $p$ = 0.1825] (Fig. 2c). In line with anterior EC, the anterior subiculum contained information regarding the allocentric goal direction [non-parametric Monte Carlo significance test: $p$ = 0.0176] (Fig. 2d). It was not possible in this region, however, to decode allocentric boundary direction [non-parametric Monte Carlo significance test: $p$ = 0.1197]. Given that the posterior and anterior sections of both the EC and subiculum lie adjacent to one another, it is possible that decoding scores in individual ROIs reflect the leakage of information between neighbouring regions. To test for this, we carried out additional control analyses in which we eroded the masks to reduce the potential influence of adjacent cortical regions (Supplementary Note 2). As can be seen in Supplementary Fig. 2, the results remained relatively consistent even when removing these additional neighbouring voxels.

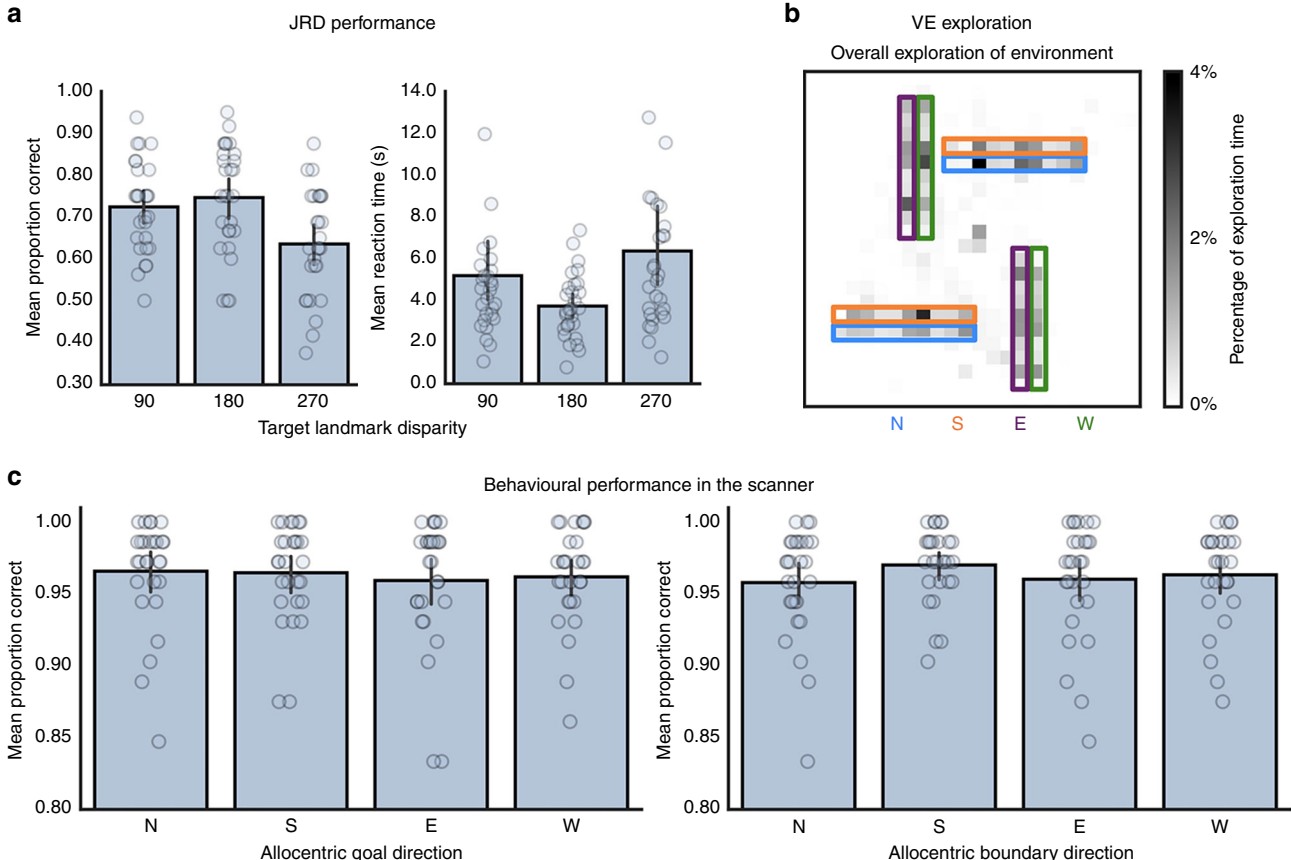

**Fig. 1** Mean performance during the learning phase and fMRI scan task. **a** On the JRD task, participants made more accurate judgements regarding landmarks located either at 180°, or 90° angular disparity, and made the fastest responses for landmarks located at 180° disparity. **b** Exploration time in the environment was matched across the four different allocentric boundary directions. **c** In the fMRI scan task, accuracy was matched for allocentric goal direction, and the same was true when grouping the trials according to allocentric boundary direction. Individual subject's data points are represented by grey circles. Error bars represent the 95% CI. Source data are provided as a Source Data file

Outside of our EC and subiculum ROIs, we were able to decode allocentric goal direction in the PHC [non-parametric Monte Carlo significance test: $p = 0.0064$]; decoding of allocentric boundary and goal direction did not survive Bonferroni-correction in either the CA1, or CA23/DG (Supplementary Fig. 3). Repeating the analysis in which we eroded the ROIs, however, resulted in these PHC effects no longer being significant (see Supplementary Note 2 and Supplementary Fig. 4). Control analyses in which different portions of the trial after movement onset were used for the decoding analysis suggest that the effects observed in EC and subiculum were not driven by visual information (Supplementary Note 3 and Supplementary Fig. 5). Furthermore, testing for egocentric boundary or goal direction coding revealed evidence only of an egocentric goal direction response in the anterior subiculum of the EC and subiculum ROIs (Supplementary Note 4 and Supplementary Figs. 6 and 7).

## Discussion

In the current study, we provide the first evidence that brain regions, analogous anatomically to those in the rodent brain, the posterior EC and subiculum, code for the allocentric direction to environment boundaries. Moreover, we found that anterior sections of these structures code for the allocentric goal direction. These data support the notion of a division of labour in the EC, in which different regions support processes involved in spatial navigation, and the coding of external sensory information[8]. Our findings are broadly consistent also with previous research in

humans that has shown functional differences in the EC according to stimulus type (i.e., scenes versus objects)[6,7].

Environment boundaries support successful navigation by providing an error correction signal when navigation is based upon path integration[12], static positional information during landmark navigation[23], and provide strong cues for reorientation[13]. Consistent with the rodent literature, we were able to decode allocentric boundary direction in posterior EC and subiculum. Although previous studies have shown univariate responses associated with boundaries[24,25], by using multivariate analysis methods, we have provided the first evidence that this medial temporal lobe boundary signal contains also the allocentric information crucial for both grid and place cell function. Moreover, in contrast to previous research, we were able to separate the contribution of allocentric boundary and allocentric goal coding[26]. We observed above chance decoding of allocentric boundary direction also in anterior EC, albeit with lower accuracy than that observed in posterior EC[30,31]. Although border cells are more prominent in the rodent MEC versus LEC, our data is in line with a weak boundary signal that has been reported in rodent LEC[31], and may be explained by the transmission of information between the two regions, due to their high levels of connectivity. Given that we see the same pattern of data in posterior EC and subiculum, it might lead to questions as to whether this represents a redundancy of function, with the same information represented in both regions. One possible difference between the posterior EC and subiculum might be information regarding the distance to the environment boundary. Although we manipulated

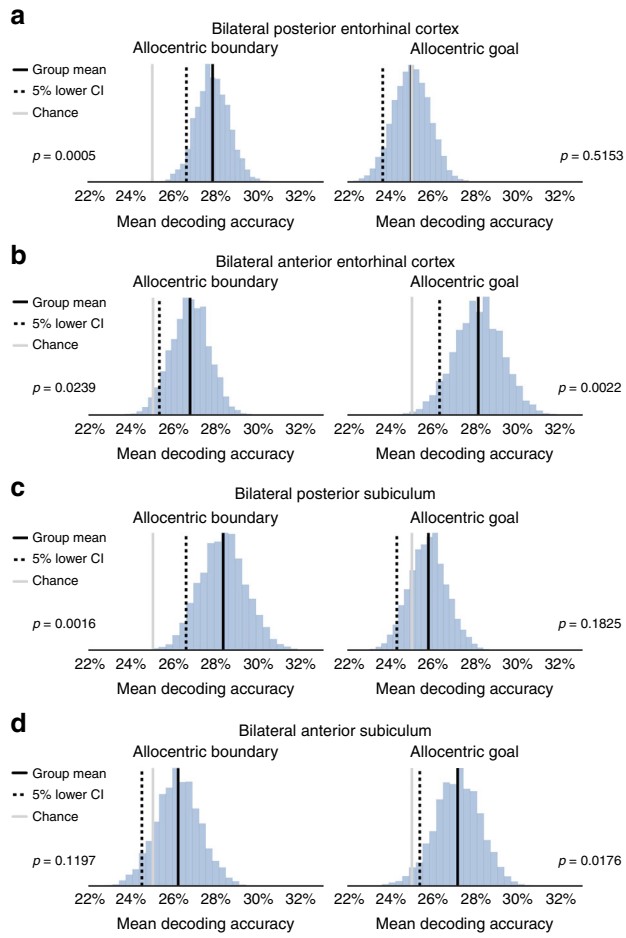

**Fig. 2** fMRI decoding results in bilateral EC and subiculum. **a** Using a linear SVC with L2 regularisation, in bilateral posterior EC group decoding accuracy for allocentric boundary direction was significantly above chance, as determined by a boot-strap permutation test (10,000 samples). It was not possible, however, to decode allocentric goal direction. **b** In anterior EC, we were able to decode both allocentric boundary and goal direction. **c** The posterior subiculum contained similar information to the posterior EC, in that we could decode allocentric boundary, but not goal, direction. **d** Finally, in anterior subiculum it was possible to decode allocentric goal, but not boundary, direction. All p values were determined via non-parametric Monte Carlo significance tests. Source data are provided as a Source Data file

only allocentric boundary direction in the current study, a key component for boundary vector cells is that they code also for distance to a boundary[18], with evidence of this coding coming from recordings from the subiculum of intracranial implant patients[25]. Examining the sensitivity of different brain regions to boundary proximity remains an important future question for boundary coding research, and the subiculum would be a likely candidate to contain this information.

A recent study discovered object–vector cells in the rodent MEC that show an allocentric directional response to objects within the environment[32]. Given that both boundary and object-related responses are evident in MEC, and that object–vector cells respond also to boundary-like structures (i.e., elongated objects), it is not possible to distinguish whether our effects are driven by border and/or object–vector cells. Consistent with previous definitions of boundaries, in the current study participants had experience that the walls impeded movement[10,18,32], and they comprised an extended 3D surface[33]. Moreover, to our

knowledge, there are no reports of object-related firing in the rodent dorsal subiculum. While it is possible that object–vector cells contribute to the posterior EC effect observed in the current study, a more parsimonious explanation is that the walls were considered boundaries and that the decoding performance reflects border and boundary vector cell responses in the EC and subiculum, respectively. The distinction between object–vector and boundary responses in the EC awaits further clarification in humans.

In contrast to MEC, the rodent LEC receives direct projections from the object-sensitive perirhinal cortex, and it has been hypothesised that it codes for external sensory information, such as prominent objects in specific locations that may constitute landmarks[8]. Consistent with this interpretation, we observed above chance decoding of allocentric goal direction in anterior EC and subiculum. Our data support previous studies in which increased activity in anterior hippocampus is associated with successfully navigating to a goal location[34]. It should be noted, however, that our participants never actually visited the goal location which could have prevented the formation of a stable goal representation supported by, for example, hippocampal CA1[35]. Evidence from rodent studies suggests that not only does the rodent LEC code for objects in specific places, but that it is more responsive to local cues rather than distal landmarks[36]. Specifically, when two sets of cues (local versus distal) were placed in opposition, the population response of LEC neurons tracked changes to the local cues. In the current study we did not manipulate global versus local features, but it is conceivable that the "global versus local" division of labour in EC emerges when there are multiple reference frames that need to be coordinated. Future studies in humans will be required to test whether the EC differentially codes for these different spatial cues.

Previous studies of EC function in humans have supported a division of labour according to object versus scene/spatial stimuli[6,7]. Why this distinction emerges according to stimulus type, however, remains unclear. One possibility is that, due to foveal vision, primates visually explore space, which in turn engages neural populations that support spatial navigation. For example, grid-cell activity has been demonstrated during the passive viewing of scene stimuli[37], with evidence also of neurons with a profile similar to that of border cells. Similar data have been reported recently in human fMRI whilst participants visually explored 2D scenes[38]. The scene versus object distinction, therefore, may reflect the engagement of these spatially tuned neural populations during the viewing of scene stimuli. The current study helps elucidate further the exact mechanism underlying this stimulus-specific effect in the EC, and suggests that, in part, the processing of boundary information in scenes drives this scene-sensitive effect in posterior EC. In contrast, recent human studies demonstrate that objects are more likely to engage the anterior EC, which is closely connected with perirhinal cortex. The current data are not inconsistent with this notion, as the allocentric goal direction signal could be interpreted as reflecting the participant bringing to mind a specific landmark object. It was not possible to disentangle the allocentric goal/landmark effect in the current paradigm because we maintained the same configuration of landmarks so that participants had a coherent understanding of the layout of the environment. Given the sparsity of the VE, changing the configuration of the landmarks during the experiment would most likely have confused participants resulting in incorrect responses and therefore lost trials. Understanding the precise role of the anterior EC, and what manipulations govern its involvement during spatial tasks, remains an important clinical objective, given, for example, that proteins such as tau aggregate in the transentorhinal area (comprising perirhinal cortex and anterior EC) early in Alzheimer's disease[39].

Outside of our key ROI, we observed significant decoding of allocentric goal direction in the PHC. These findings are in line with other data that suggest the PHC contains information regarding the direction to an imagined goal[40]. Specifically, when participants were required to recall the direction between two well-learned goal locations only the PHC was sensitive to similarities in imagined direction. These data are compatible with the findings reported here for a task in which participants were required to remember the allocentric direction to a goal landmark given the current heading direction. Alternatively, this above chance decoding could reflect the participants bringing to mind the specific landmark, with landmarks also known to engage the PHC[41]. Caution should be expressed when interpreting these results, however, given that the effects were no longer significant when using eroded masks.

Although we did not find evidence that the PHC is involved in the processing of allocentric boundary direction, this region has been shown to be exquisitely sensitive to scene stimuli, and in particular the structure of a scene[41,42]. Scene-selective portions of the PHC discriminate scene stimuli depending upon whether they contain highly visible boundaries regardless of scene content[43], and add to a network of brain regions including V1 and the lateral occipital cortex that are sensitive to boundary information in scenes[44]. Furthermore, the occipital place area has been shown to be causally involved in memory for object locations relative to boundaries but not landmarks[21]. It remains to be explored in humans how this lower-level visual information regarding scene structure is combined with the allocentric representations necessary to support allocentric boundary direction coding observed here in the EC and subiculum. One possibility is that representations of these boundary features in the environment are combined, via conjunctive neurons, with both head direction information and egocentric positional estimates relative to the environment walls, as has been demonstrated in rodents[45].

Scene-specific responses have been reported also in the human anterior subiculum[46]. Although these data may seem at odds with our posterior subiculum boundary effects, it is possible that anterior subiculum shows a univariate scene response, whereas the multivariate pattern in posterior subiculum is informative of allocentric boundary information in the absence of greater scene-related activity. Moreover, while an alternate multivariate analysis strategy—representational similarity analysis (RSA)—supported the decoding results of allocentric boundary coding in posterior EC, the effect in posterior subiculum was not significant (see Supplementary Note 5 and Supplementary Table 1). The discrepancy between the decoding and RSA allocentric boundary results in the subiculum may reflect the fact that the linear SVC uses only the most informative voxels to form a decision hyper-plane between categories, whereas the RSA tested for the degree of similarity across all voxels in the ROI. These differences between subiculum and EC, therefore, may speak to a sparser voxel-level representation of allocentric boundary direction across regions. Alternatively, representations of different allocentric boundary directions might be less distinct in the subiculum while still allowing for successful decoding. Future studies will be necessary to elucidate the nature of scene-sensitivity in the subiculum, and the precise perceptual features driving these effects.

Neuronal populations originally thought to support only spatial navigation have been shown to be involved also in more abstract cognitive processes. For example, in rodents MEC grid cells map not only space, but also different sound frequencies[47]. Similarly, grid cell-like representations in humans, revealed via fMRI, have been found to support the organisation of conceptual knowledge[48]. In the current experiment, therefore, although we have used a very concrete example of a boundary coding (i.e., a physical boundary), the same posterior EC mechanism may

support more abstract boundary-related processes, such as the segmentation of temporal information into event episodes, or the coding of visual boundaries[49]. There is evidence that boundaries are used to segment a continuous temporal stream into distinct episodic events. For example, increased forgetting of object pairs is observed when having to remember items between-rooms versus within the same room[50]. Furthermore, activity in the hippocampus has been shown to correlate with the salience of event boundaries during the viewing of films[51]. The ERC and subiculum, therefore, may also play a critical role in the formation of event episodes.

Taken together, our study provides the first evidence of allocentric boundary coding in humans, and suggests that, consistent with models of anatomical connectivity, posterior EC and subiculum provide support for positional coding, whereas the anterior EC and subiculum code for external sensory information such as landmarks. These findings advance our understanding of EC function, and provide further mechanistic explanation underlying the division of labour in this region.

## Methods

**Participants.** In total, 31 right-handed, young healthy adults (13 female; mean age 26.12 years, range: 20–33 years) participated in the experiment and were paid 31 euros for their time. All participants provided informed consent, and the experiment received approval from the Ethics Committee of the University of Magdeburg.

**General procedure.** The experiment comprised two separate days of testing. On the first day, the participant learned the layout of a VE outside of the scanner. On the following day of testing, the participant underwent high-resolution fMRI scanning.

**The VE.** The VE was created using WorldViz Vizard 5.1 Virtual Reality Software (WorldViz LLC, http://www.worldviz.com). It comprised a large grass plain (600 × 600 virtual metres[2]; invisible walls that prevented the participant from leaving the VE resulted in a 500 × 500 virtual metres[2] explorable area) surrounded by four distinct global landmark cues, and contained four rectangular boundaries (2 × 4 × 40 virtual metres), each with a unique brick texture (Fig. 3a; for rights reasons, the images used here for the Figures differ to those used in the experiment). The global landmark cues (a mountain, a tower, a cathedral and a city) were rendered at infinity, and indicated cardinal heading directions in the environment. During the experiment, however, the landmarks were never referred to using cardinal directions (i.e., north, south, east and west). In the VE, two of the four boundaries were arranged with their long axes spanning north-to-south, whereas the other two boundaries spanned east-to-west.

**Training outside of the scanner.** A head mounted display (HMD; Oculus Rift Development Kit 2) was used during training to provide an immersive learning experience. The participant stood during the experiment and was required to physically turn on the spot to change facing direction in the VE; translations were controlled via a button press on a three-button wireless mouse held in the participant's right hand throughout the training phase. To promote exploration of the VE and its boundaries, the participant was required to collect blue tokens (0.25 virtual metre radius spheres positioned at a height of 0.8 virtual metres; the first-person view in the VE was rendered at 1.8 virtual metres) that formed a path around the four boundaries (Fig. 3b). The participant was required to walk through each token after which it disappeared; the participant was free to collect the tokens in any order. Furthermore, to ensure that the participant was aware that the boundary was impassable, they were required to "activate" red sensors located on each side of a boundary (eight sensors in total, positioned at a height of 2 virtual metres; wall trigger radius = 0.2 virtual metres), via a button press, which resulted in them turning green.

After all tokens had been collected, and all sensors activated, the participant completed a JRD criterion task, which was used to assess their knowledge of the VE's layout (Fig. 3c). On each trial of the JRD task, the participant was presented with a static picture of one of the four global landmarks (1 s), which they were required to imagine facing. After a brief pause (0.5 s), a picture of a different global landmark was shown and the participant was required to indicate the direction of the second landmark relative to the first. Specifically, if the participant thought that, when facing the first landmark, the second landmark was located to the participant's left then they pressed the thumb button on the mouse (i.e., the left-most button); if the second landmark was located behind them, they pressed the left mouse button (i.e., the middle of the three response buttons), and if it was located to the right, they pressed the right mouse button (i.e., the right button). Performance was assessed via the number of correct responses, with the participant

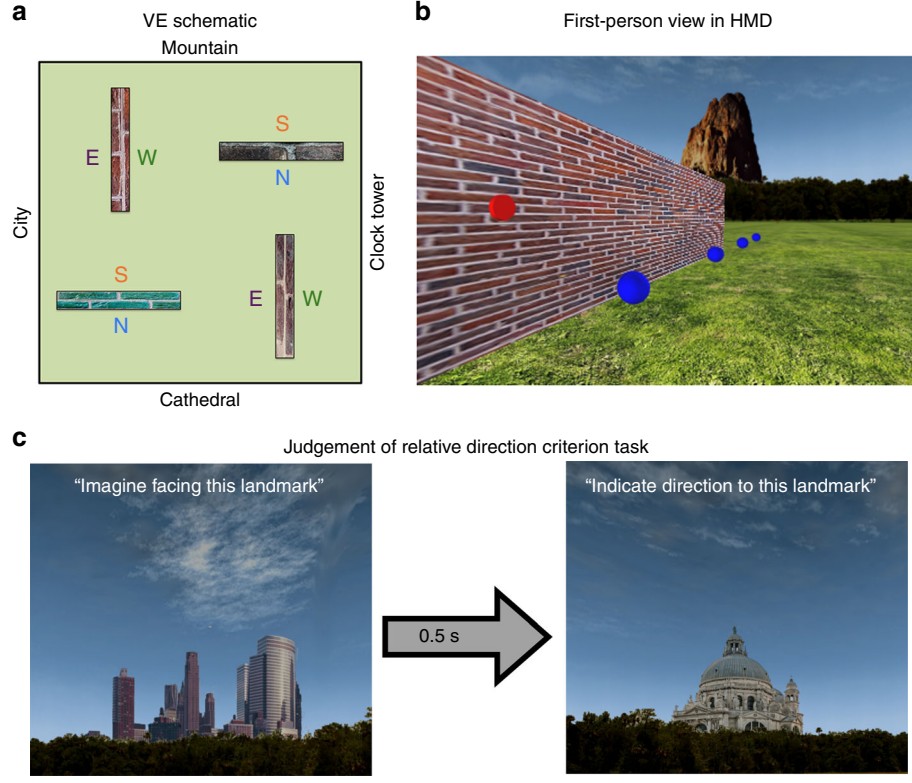

**Fig. 3** Schematic of the VE and criterion task during learning phase. **a** The VE comprised four boundaries, each with a unique texture. The long axis of two of the boundaries spanned North-South, and two spanned East–West. The VE was surrounded by four global landmarks rendered at infinity that provided information regarding cardinal direction in the environment. Each side of the boundary created an allocentric boundary either to the North (N), South (S), East (E), or West (W). **b** In the learning phase the participant wore an HMD and controlled their orientation by physically rotating on the spot whereas translations were controlled via a button press. During exploration, the participant was required to collect all blue tokens and activate all red sensors located on the different sides of the boundaries. **c** After exploring the environment, the participant completed a JRD task in which they were presented with one of the four global landmarks and asked to indicate the relative direction to another global landmark; the landmarks shown here are the City (pictured to the left) and Cathedral (pictured to the right). Participants were allowed to move on to the scanned test phase only after answering all JRD questions correctly

proceeding to the fMRI scanner task only if they answered all 12 JRD questions correctly; an incorrect response resulted in the participant returning to the VE to repeat the exploration phase.

**fMRI task**. In the fMRI task, the participant viewed passive movement in the VE and was required to indicate the global landmark located in the direction of a cue object positioned either to the left or right of their path. Each trial comprised passive movement along a predefined path in which the participant could see (1) one global landmark towards which they were moving, (2) one boundary and (3) the cue object (Fig. 4a). After the movement ended (2 s), the screen faded to black for 4 s before the start of the decision phase (2 s), which comprised a forced-choice response. Here, the participant had to indicate which of the three global landmarks (i.e., the remaining landmarks not seen during the passive movement on the trial) was located in the direction of the cue object. For example, if the participant viewed a path heading towards the mountain, and the cue object was positioned on the right-hand side of the path, the participant was required to identify the global landmark located to the right of the mountain. In this case, the correct response would be the clock tower. In the forced-choice decision, the three global landmarks were presented on screen in a row, with the position of the landmarks randomly assigned either to the left, middle, or right position of the screen on each trial; randomising the screen position-landmark associations was important to ensure that they did not confound any subsequent decoding analyses (Supplementary Note 6 and Supplementary Fig. 8). The participant had to select, via a right-hand MR-compatible button box, which of the landmarks they thought was located in the direction of the cue object using either a thumb, index, or middle finger response, corresponding to the landmark image's position on the screen. Videos of both the exploration phase and fMRI task can be found online (http://www.wolberslab.net/boundarycoding.html).

By using predefined paths in the VE we were able to control the position of the boundary and goal object relative to the participant. There were three paths per side of the boundary (24 in total), and these paths resulted in the boundary being located either to the left, right or straight in front of the participant (Fig. 4b). Each

path was repeated four times per run (96 trials per run; three runs in total) and the cue object's position changed over trials so that its position was balanced across the left and right side of the path (i.e., for each path repeated four times per run, the cue object was located twice to the right and twice to the left of the path). Trials could then be binned to examine different questions regarding allocentric boundary or allocentric goal direction coding (Fig. 4c). Importantly, these different spatial properties were balanced across the different conditions, meaning that the comparisons were orthogonal (Supplementary Fig. 9). For example, trials used to examine allocentric boundary to the North would comprise views of two different boundaries (with their distinct textures), views of different global landmarks, the egocentric location of the boundary location to the participant's left, right and front, allocentric goal locations pointing to all four global landmarks as well as an equal number trials in which the goal object was located egocentrically to the left or right of the participant. Each trial lasted 8 s with a mean 1 s inter-trial interval and each of the three runs lasted 14.8 min.

**fMRI data acquisition**. Imaging data were acquired using a 3 T SIEMENS (Erlangen, Germany) Magnetom Prisma scanner, with a 64-channel phased array head coil. Scans comprised a whole-head, three-dimensional structural T1-weighted anatomical image with 1 mm isotropic resolution (repetition time (TR)/echo time (TE)/inversion time = 2500/2.82/1100 ms; flip angle = 7°; field of view (FOV) = 256 × 256 mm; 192 slices; GRAPPA acceleration factor 2); a high-resolution moderately T2-weighted structural image comprising the hippocampus and EC acquired perpendicular to the long axis of the hippocampus using a turbo-spin-echo sequence (in-plane resolution = 0.4 × 0.4 mm, slice-thickness = 1.5 mm; TR/TE = 4540 ms/44 ms; FOV = 224 × 224 mm; 32 slices); gradient echo field maps (in-plane resolution = 1.6 × 1.6 mm; slice-thickness = 2 mm; TR/TE1/TE2 = 720/4.92/7.38 ms; flip angle = 60°; FOV = 220 × 220 mm; 72 slices) and three runs (445 volumes each) of T2*-weighted functional images acquired with a partial-volume echo-planar imaging sequence, aligned with the long axis of the hippo-campus (in-plane resolution = 1.5 × 1.5 mm, slice-thickness = 1.5 mm + 10% gap;

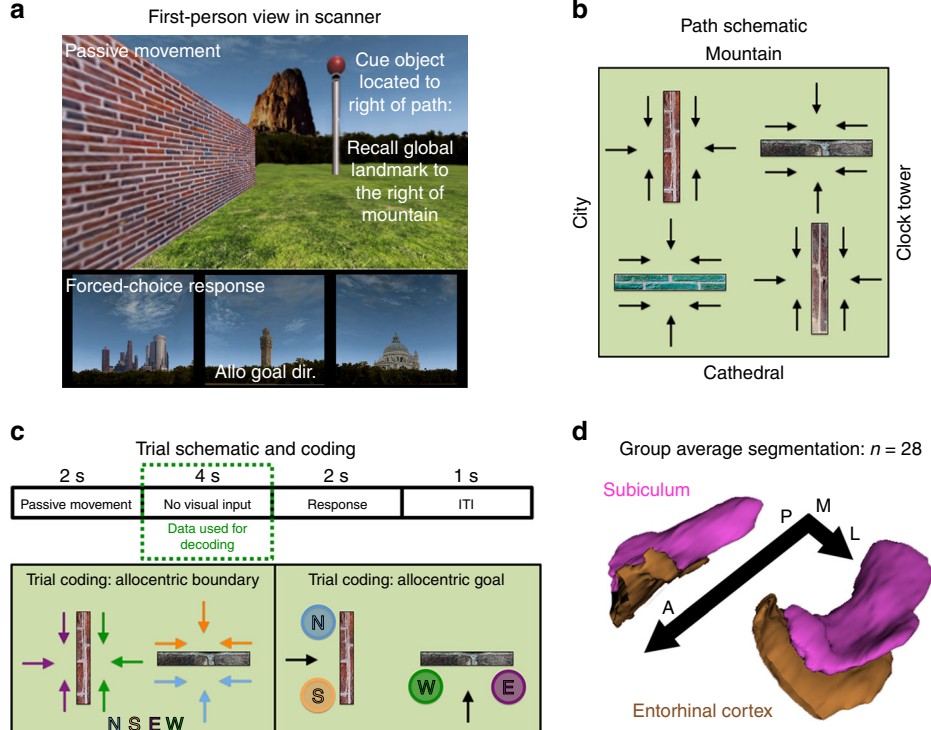

**Fig. 4** fMRI scanner task, trial coding and example group ROIs. **a** On each scan trial, the participant viewed the global landmark towards which they were travelling, a boundary and a cue object (pole with ball). The participant then completed a forced-choice decision in which they were presented with the three other global landmarks (i.e., those not seen during the passive movement) and were required to select the landmark located in the direction of the goal object (correct answer here indicated by "Allo Goal Dir"). **b** Schematic of the 24 passive paths used in the scan task. Each path was repeated four times resulting in 96 trials per run. **c** Each trial comprised 2 s of passive movement, four seconds in which the visual input was removed (i.e., the portion of the trial used for the fMRI analysis) and 2 s to make the cue object direction decision. Trials were then coded according to either the allocentric boundary direction or the allocentric goal direction. **d** All analyses were carried out in the participant's native EPI space using manually segmented masks of the EC and subiculum; the group averaged ROIs are presented here for display purposes only

TR/TE = 2000/30 ms; flip angle = 90°; FOV = 192 × 192 mm; 26 slices; GRAPPA acceleration factor 2).

**fMRI data preprocessing**. A custom preprocessing pipeline was created using Nipype[52], in which we combined packages from SPM12[53], FSL5[54] and Advanced Normalisation Tools[55] (ANTS 2.1). The pipeline comprised realignment of the EPI data to the first volume of the series (SPM), intensity-normalisation (FSL) and high-pass filtering with a cut-off of 128 s (FSL). Structural T1 images were bias-corrected (SPM) and segmented (SPM), with the resulting grey matter, white matter and CSF tissue probability maps combined to create a mask for brain extraction.

Using FSL's epireg, EPI data were coregistered to the structural T1, whilst also applying field map correction to the functional images using field maps acquired during scan sessions. High-resolution T2 images were coregistered to the T1 using ANTS. Manually segmented hippocampal T2 ROIs were coregistered to the EPI data by concatenating the T2-to-T1, and EPI-to-T1-inverse matrices using ANTS (Supplementary Note 6 and Supplementary Fig. 10). The EPI-to-T1-inverse matrix was used to move the T1 brain mask (comprising grey matter, white matter and CSF) into EPI space. The EPI data were then multiplied by this brain mask to remove all non-brain tissue. In order to maintain as accurate anatomical specificity as possible, all subsequent analyses were conducted on unsmoothed EPI data.

**Medial temporal lobe masks**. Bilateral hippocampi and parahippocampal cortices were segmented manually on the individual subjects' T2 images using "ITK-SNAP"[56], following an established protocol[57] (Fig. 4d). Given the differences in connectivity along the anterior and posterior portions of the EC and subiculum, we split each of the individual participant's ROIs in the middle of the long axis, separately for each hemisphere (see Supplementary Fig. 11 for size and temporal signal-to-noise ratio of the ROIs). Due to movement artifacts in the T2 images, it was not possible to segment the hippocampi of three male participants. Consequently, all results reflect the data from 28 participants (13 females). For a separate control analysis to assess the possible effects of leakage of information between neighbouring ROIs, we created more conservative versions of our masks in which we eroded voxels on the perimeter of the ROI using the erode function in FSL maths (box kernel = 1 × 3 × 1 voxels) (Supplementary Fig. 12).

**Data analysis**. Prior to decoding analysis, movement parameters obtained from the realignment of the functional images were regressed out of the data[40]. Here, we included 24 regressors in the model, reflecting the realignment parameters, their derivatives, their squares and their square derivatives[58].

Each of the 96 trials per run was modelled separately in the analysis. To reduce the possible influence of visual information in our decoding analysis, we analysed the portion of data corresponding to the period of the trial after the passive movement ended during which there was no visual input (i.e., a black screen) and was therefore matched across different allocentric boundary/goal directions. Given its high performance in decoding event-related functional imaging data with short inter-stimulus intervals, the "Add"[59] model was implemented here. This model aims to capture the putative peak of the haemodynamic response function occurring 4–6 s after the onset of the event of interest. Since we wanted to capture activity associated with the stationary period of the trial, which occupied the period 2–6 s after trial onset (Fig. 4c), we took the estimates from an unconvolved boxcar regressor that spanned three TRs occurring 4–6 s after the stationary phase[40,60] (i.e., 6–12 s after trial onset), in separate models comprising one regressor representing the trial of interest, and a second regressor modelling all other trials in the scan run. Consistent with previous studies we also averaged the estimates over separate runs to boost the signal-to-noise ratio[61]. Specifically, to enhance the signal corresponding to the allocentric condition of interest whilst maintaining the voxel space, we created an average over the three runs by first ordering the trials in each run according to the condition to-be-decoded (allocentric boundary or allocentric goal). The rationale here was to strengthen the condition of interest, whilst weakening any signal associated with other conditions (e.g., head direction). This trial-averaging resulted in 96 samples per participant, balanced equally across North, South, East and West directions for allocentric boundary and allocentric goal conditions.

A linear support vector classifier (SVC) with L2 regularisation as implemented in Scikit-learn[62] was used for the decoding of different allocentric directions. The regularisation strength was determined by adjusting the C hyperparameter. In order to follow decoding best practices[63], we used nested cross-validation to estimate the best C hyperparameter and obtained a cross-validated estimate of the classifier accuracy with three outer folds using 20% of the data as test set in each fold. The best hyperparameter was chosen within the inner nested cross-validated fold, using a grid search with possible values in the range of $1$–$10^3$ in steps of power

of 10. All decoding analyses were conducted in the participants' native EPI space (see Supplementary Fig. 13 for an overview of the analysis pipeline).

**Statistical tests**. All behavioural data (mean accuracy and RT data from the learning phase JRD and fMRI task) were submitted to repeated-measures ANOVAs calculated using SPSS (IBM Corp. Version 21.0). Mauchly's test of sphericity was used to assess homogeneity of variance for the ANOVAs, and Greenhouse–Geisser estimates of sphericity used to correct degrees of freedom when this assumption was violated. Given that we had no apriori predictions as to differences in performance across the different conditions of the behavioural tasks, follow-up two-sided paired-sample $t$ tests interrogating significant main effects and/or interactions were Bonferroni-corrected for multiple comparisons. Effect sizes were calculated using online tools[64], and all plots were created using a combination of Matplotlib[65] and Seaborn[66].

For the decoding analyses in the separate ROIs, we obtained the mean decoding score per participant over the three-folds of the cross-validation. We then used the bias-corrected and accelerated boot-strap[67] (BCa) to sample from these values 10,000 times to obtain the distribution of our group-level decoding accuracy[68]. Non-parametric Monte Carlo significance testing[69,70] was used to generate a $p$ value based on the distribution of our data, where we first subtracted the group-level decoding accuracy from each participant's decoding score, before adding chance performance (i.e., 25%). This had the effect of shifting the distribution of our group's decoding scores to around chance performance, and we then again used the BCa (with 10,000 samples) with these values to generate our null distribution. The one-tailed $p$ value was calculated by counting the number of times the boot-strap null mean exceeded our observed group-level decoding score and dividing this value by the number of samples (i.e., 10,000); importantly, 1 was added to both the numerator and denominator of this calculation to correct for cases where none of the boot-strap null means exceeded the group-level decoding score. Outside of our key ROIs (EC and subiculum) we tested also whether we could decode allocentric boundary and goal direction in manually segmented masks of the CA1, CA23/DG and PHC. Given that we did not have overt predictions as to the expected pattern of results in these ROIs, we used a Bonferroni-adjusted alpha level to test for significant effects (three ROIs × two conditions = 0.05/6 = 0.008 adjusted alpha).

**Reporting summary**. Further information on research design is available in the Nature Research Reporting Summary linked to this article.

## Data availability

The data that support the findings of this study are available from the corresponding author upon reasonable request. The source data underlying Figs. 1a–c and 2, and Supplementary Figs. 1–8 and 11 are provided as a Source Data file.

## Code availability

The custom code used to analyse the data are available from the corresponding author upon reasonable request

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

## Acknowledgements

We would like to thank Rebecca Korn for help with data collection, Arturo Cardenas-Blanco for support in functional image processing, Paula Vieweg and David Berron for guidance in hippocampal segmentation and Matthias Stangl for helpful discussion. This research was funded by an ERC Starting Grant AGESPACE (335090) awarded to Prof. Dr. Thomas Wolbers.

## Author contributions

J.S., J.V.H. and T.W. designed the experiment; C.T. created the scanner sequences; J.S. and J.V.H. collected and analysed the data; J.S., J.V.H. and T.W. wrote the paper.

## Additional information

**Competing interests:** The authors declare that they have no conflict of interest.

