## [Transparent Peer Review File · Nature Communications]

Response to Reviewers

Reviewer #1 (Remarks to the Author):

The manuscript presents an fMRI study in which participants first learned the layout of a virtual reality (VR) world (presented in an immersive VR) containing 4 non-connected walls in a green space surrounded by nameable features projected at infinity (e.g. mountain, cathedral) providing cardinal direction cues. To proceed to scanning, participants need to make judgments of relative direction to a cued object based on an initial heading. E.g. you are looking at the mountain and which direction is the cathedral? During scanning participants watched short linear motion in the environment for 2 sec with now cue objects added (red ball on a stick). After 4 seconds of a black screen a set of 3 of the environmental features appeared and the participant had 2sec to judge which one of these lay in the direction of the ball on the stick. e.g. if they saw travel towards the mountain and the ball on the stick was on the right the correct (goal) answer was the cathedral. The experimental design was such that participants could move along a set of trajectories that independently sampled the allocentric direction to the walls (e.g. having a boundary to the N,S,E,W) and the allocentric direction to goals. Using a linear vector support machine to decode the boundary and goal information the authors report a double dissociation where posterior EC/Sub regions decode boundary information, whereas anterior EC/Sub decode goal information. The authors interpret this data in relation to rodent data.

This experiment tackles an important set of questions and uses a nice design to get around the problems in past experiments where boundary and goal direction were confounded. The high-res fMRI and bootstrapping approach for the analysis is a nice feature of the study. The results are interesting and have the potential to advance current models of navigation.

I have a number of concerns the authors I hope can address to improve the manuscript.

1. The authors report that RTs differ for the different goal directions (Fig S4)? Thus, it is possible that the ease of retrieval of the goals might be what the LSV is decoding rather than goal direction. I wondered if the authors could address

this by exploring whether when sub-sampling the data to compare decoding across conditions. If it is ease of decoding it should be easier to decode N from S rather than E from W for example. Showing the decoding does not follow the RT pattern would be useful to address this issue.

We thank the Reviewer for raising this important point as to whether the linear SVM was sensitive to differences in RT. To address this question, we carried out six binary classification models (i.e., North versus South (NvS), North versus East (NvE), North versus West (NvW) etc.), for each ROI. The resulting decoding accuracies were submitted to a repeated-measures ANOVA comprising the factors ROI (EC, subiculum) \times Portion (Anterior, Posterior) \times Model (NvS, NvE, NvW, SvE, SvW, EvW), which did not reveal any significant main effects or interactions (all $F_s < 2.13$, $p_s > 0.15$). If the classifier were detecting differences in RT then the main effect of Model would be of particular importance, however there was no evidence that this factor modulated decoding accuracy ($F(5, 135) = 0.94$, $p = 0.46$). To test more thoroughly any ROI-specific effects, pair-wise T-tests were carried out for the models most likely to show an effect driven by RT. Specifically, in line with the RT data, one might predict that the decoding accuracy of North versus West should be greater than for North versus South, given that North and West show the greatest disparity in RT. Consistent with the ANOVA, however, there was no evidence that decoding accuracy was higher for NvW relative to NvS in any of the EC or subiculum ROIs (all $t_s < 1.08$, $p_s > 0.58$).

These analyses have now been included in the Supplementary Information.

2. Because the time-line was fixed (always 4 sec from the movie to judgment) it seems not possible to rule out the influence of the visual information on the decoding analysis. It would be useful if the authors could conduct an analysis the visual content presented in each of the conditions to convince readers that the decoding was unlikely to be driven by visual information. e.g. by sub-sampling the data in some manner to explore the similarities.

The Reviewer is correct that it is difficult to rule out the influence of visual information, but we would first like to clarify that the trial design (i.e., visual input) and averaging that we implemented was balanced across different spatial properties (e.g., cue object location, egocentric boundary position). As a consequence, our design ensured that

there were no systematic differences in visual input between, for example, the four allocentric boundary directions.

In addition, to test whether the decoding in EC or subiculum was driven by perceptual features of the stimuli, we have conducted further analyses. Firstly, by selecting different portions of the trial for our decoding analysis, we sub-sampled the data to examine the temporal evolution of the boundary signal. Our hypothesis was that if the passive movement had induced a visual confound, the TRs encompassing the peak of the haemodynamic response function (HRF) should provide maximum information about the visual content. As a consequence, TRs around 4-6 seconds after the onset of passive movement should show higher decoding scores relative to later time windows when the HRF is nearing baseline again. As can be seen in Supplementary Figure 13, in the posterior EC and subiculum, the highest decoding accuracies were achieved later in the trial, and not in the TRs corresponding to the portion of the trial encompassing 4-6 seconds after the viewing of the passive movement where visual content would likely have its strongest influence (i.e., 2-8s). Secondly, in response to your comment number 4 regarding the decoding of egocentric conditions, we split the trials according to egocentric boundary direction. This analysis implements a strong visual confound, because the boundary is classified according to its position in the visual field (i.e. right, left or centre). As expected, this visual confound made it possible to decode egocentric boundary information in area V1 (Supplementary Figure 12). In contrast, we did not observe above-chance decoding in either the posterior EC or the subiculum (Supplementary Figure 11).

Together, these findings strongly suggest that it is unlikely that low-level visual features might have driven the decoding of allocentric boundary / goal directions in EC or the subiculum. Importantly, our results are also in line with previous research that has used elegant controls to test for the coding of visual information in medial temporal lobe regions. For example, Chadwick et al. (2015) tried to decode distinct visual information from their virtual environment in their EC/subiculum cluster and found that although this region was sensitive to directional information, it did not code for perceptual qualities of the stimuli.

Supplementary Figure 13. *Group-level decoding accuracy for allocentric boundary and allocentric goal direction as a function of time after trial onset. The horizontal grey line represents chance performance (25%) and the data points surrounded by the green ticked line represent those used in the analysis in the paper using the "Add 4-6" GLM.*

3. The design appears to allow egocentric directions to be decorrelated from the allocentric directions, however it would be helpful for the authors to show this visualised in set of matrices in supplemental.

We thank the Reviewer for this comment and we have now created a matrix of the different trials and coded them according to the decoding conditions (Supplementary Figure 2).

Supplementary Figure 2. Schematic of the different trials in the scanner task and the allocentric boundary and goal direction trial coding schemes. Only two (one horizontal, one vertical) of the four boundaries are displayed given that this scheme is identical for the other two boundaries.

4. Given recent interest in egocentric boundary information and egocentric direction to goal locations in rodent and bat studies respectively, it would be very useful for the authors to add an analysis of the egocentric parameters in supplemental. Past work would perhaps predict anterior EC might contain egocentric boundary information and CA1 egocentric goal direction information.

I assume the authors have not done these analyses due to strong visual and button choice confounds, but nonetheless, the results would be useful evidence for future studies that might control for such information.

We thank the Reviewer for highlighting the importance of demonstrating evidence of egocentric coding and for completeness we have now provided in the Supplementary Information the decoding results for the egocentric boundary and egocentric goal direction conditions in the same ROIs. As the Reviewer notes, one should express caution when interpreting these results given the visual confounds due to lower-level visual similarities when averaging the trials in this way. For example, unlike in the allocentric condition, in the egocentric boundary condition the boundary will be located either to the left, right, or straight in front of the participant, which provides very distinct visual information as to the different classes of stimuli. For the egocentric goal direction, although the boundary position will be balanced across the two stimulus classes (goal cue object left versus goal cue object right), the lower-level feature of the cue object position will consistently discriminate the two conditions.

In our key ROIs, it was possible to decode egocentric goal direction in anterior subiculum only (Supplementary Figure 11). As can be seen in Supplementary Figure 12, we were able to decode egocentric boundary direction in the CA1 and CA23DG, whereas egocentric goal direction information was contained in CA23DG. Consistent with the Reviewer's intuition regarding visual confounds, we achieved high decoding accuracy for both egocentric boundary and egocentric goal direction in V1. As noted above, however, despite these strong visual confounds in the egocentric boundary condition, it was not possible to decode this information in the posterior EC or subiculum. This is consistent with the analysis presented above with regard to the inability to decode visual information in EC/subiculum.

Supplementary Figure 11. Mean decoding performance for egocentric conditions in the EC and subiculum ROIs. Chance performance is 33% for egocentric boundary direction (left, right, straight) and 50% for egocentric goal direction (left, right).

Supplementary Figure 12. Mean decoding performance for egocentric conditions in the medial temporal lobe and V1 ROIs. Chance performance is 33% for egocentric boundary direction (left, right, straight) and 50% for egocentric goal direction (left, right).

5. Fig 2 is confusing. In a. it is important to re-state the conditions for making the judgement. E.g. that the correct answer relates to the fact the cue object is

to one side of the global landmark. In c. the wall and red dot come across as a bit odd and distracting in the time-line. Seemed to me they make this more confusing than help. Most importantly the trial coding diagram leaves the reader unclear as to the full coding set up. In the right hand allocentric goal panel only two trials are shown (presumably to avoid the over-load of presenting all of them), but in each case why are 2 goals possible rather than 3 since in Fig 2a, 3 images are shown?

We thank the Reviewer for bringing to our attention that Figure 2 could more clearly explain the task and trial coding. We have now included text in the Figure to describe the conditions used to make the allocentric goal direction judgement. We have now also removed the wall and the red dot from the timeline and have included links to videos of the trials (<https://tinyurl.com/y3otgq7w>). The reason that only two goals are demonstrated in the goal trial coding panel is because the goal object was located either to the right or the left of the path. There were, therefore, only two goal directions for which this trial could be coded. During the response phase (Figure 2a), however, we provided the option to respond to any of the three landmarks not visible in the virtual environment during the passive movement.

Figure 2. Updated figure with increased information regarding the task (a), and with new trial schematic information (c).

6. On line 366 the authors describe the method for determining the p-values. It would be useful to reference another article using this method. It seems sensible, but would be useful to know who has taken a similar approach before with fMRI datasets.

We thank the Reviewer for raising this point. We used a Monte Carlo test procedure to determine the significance of our decoding accuracy (Besag, 1992; Hope, 1968). We are unaware of this method being used with other fMRI datasets, but the bootstrap methods in which the rank of the observed data is compared to the null distribution make no assumptions as to the distribution of the data. In this respect, they can be more appropriate than comparing accuracies using, for example, a T-test against chance performance. We repeated the same analyses using the non-parametric Wilcoxon Sign-test and the effects were identical for allocentric boundary decoding, and almost perfectly replicated for the allocentric goal direction, with the exceptions of the anterior subiculum ($p = 0.08$) and CA23DG ($p = 0.12$).

7. When considering how the data here relates to previous studies it would worth considering that the goal locations were never actually visited and walked around. It is unclear what impact this has, but it may have some.

We thank the Reviewer for raising this point and have now added to the Discussion to clarify that the goal direction representations observed in the current study may differ to those studies in which the subject actually visited the goal location.

Discussion (Lines: 639-642): "It should be noted, however, that our participants never actually visited the goal location which could have prevented the formation of a stable goal representation supported by, for example, hippocampal CA1⁵⁴⁻⁵⁶".

Reviewer #2 (Remarks to the Author):

Evidence for allocentric boundary and goal direction information in the human entorhinal cortex and subiculum

This study used a VR navigation training task in combination with a fMRI pattern analysis during a relative direction judgment task using passive observation of on-screen movement (through various paths in the same virtual environment).

The main regions of interest were EC and Subiculum (Sub), with control areas CA1, CA23DG, and PHG. The four “directions” analyzed were with respect to four distal landmarks (e.g., mountains, buildings) and four freestanding boundaries within the environment (two walls oriented along the axis defined by two opposing distal landmark, and the other two walls oriented 90-degrees from the first two, along the axis defined by the other two opposing landmarks). The main claim is that there is information about the allocentric direction of the boundaries with respect to the subjects and that there is information about the allocentric direction of the “goals” (cued by an object on the screen), but that these two signals originate from the posterior and anterior regions, respectively.

It is true that there are some conceptual replications of previous findings presented here, such as the general boundary-related activity in the human hippocampal subregions and the LEC/MEC (or anterior/posterior) distinction. However, the paper also has some notable merits as well, such as the commendable efforts to decode allocentric boundary direction and to distinguish between directions of distal landmarks and local boundaries all in one task. The task and methods are generally very well thought out and the reported effects quite strong. Nevertheless, there are several serious concerns about the paper that I have summarized below. I hope that they help the authors revise and potentially even reframe the central claim of the paper to create a stronger paper in the end:

1. Issue of boundary direction: Border/boundary cells are defined by their response to environmental boundaries such as walls. Often, these cells are directionally tuned; however, the indication of the direction of a boundary is less essential to their characterization than their representation of the proximity of boundaries. Hence, the border score used to classify EC border cells is simply a comparison between firing distance from the wall with the maximum coverage of a field of any of the walls. This means that many border/boundary cells will respond not only to one wall of a quadrilateral-shaped environment. On the other hand, the BVC model does afford directional tuning of boundary cells, but even with a distance component to this model, most subicular boundary cells activate proximally to a boundary. Given this, the allocentric directional coding discussed in this paper is slightly unclear, with respect to the analogy the

authors are making with the rodent brain. Although these issues are mentioned in the paper, it could be more directly tied to or contrasted with the study itself.

We thank the Reviewer for clarifying this point regarding the properties of boundary direction coding in the EC. In addition to the boundary vector cell model, the rationale for our study was also based on findings such as those reported by Solstad et al. (2008): in that study, 52/69 cells in MEC responded to a single wall in the environment; the remaining 17 had multiple fields. In a subsequent analysis, 12/22 border cells continued to show a boundary response to a newly inserted wall in the environment, and the direction of firing for these cells was maintained between the peripheral boundary wall, and the inserted boundary. These data support our assertion that it should be possible to observe allocentric boundary representations in EC. We have now added a sentence to the manuscript that acknowledges that a subset of border cells show directional modulation.

Background (Lines: 50-53): "In rodents, the MEC contains a number of different spatially-tuned neural populations, including border cells¹⁰ that fire when proximal to environment boundaries, with a subset of these indicating also the boundary's allocentric direction."

2. Issue of boundary specificity: There is some evidence of object vector cells in the EC. Usually, the way to distinguish a boundary vector cell from an object vector cell would be to look for an extended firing field along the boundary; this is particularly noticeable when there is an environmental wall (to which object vector cells do not fire). From the results presented here, it doesn't seem possible to distinguish whether the results are object-vector-cell-related or boundary-related. I find this to be one of the most serious issues that might put the author's claims about boundary representations at jeopardy. Could this be addressed somehow?

We thank the Reviewer for raising this valid point as to whether our reported effects can be considered boundary-specific, or if they could be considered evidence instead for object-vector cells in humans. Using fMRI, we are unfortunately not able to test for boundary/object effects in real time as can be carried out in rodent electrophysiology.

It is true that the "boundariness" of a stimulus is a continuous and abstract property (Lever, Burton, Jeewajee, O'Keefe, & Burgess, 2009), and it can be difficult to determine the point at which one might expect an object to be classed, both

physically and psychologically, as a boundary. In our virtual environment, the length of the boundary was 40 virtual meters and its height 4 virtual meters. This is over 20 times as long as the height of the participant rendered in the environment, and twice as tall. These features are consistent with the definition of a boundary being an extended 3D surface (Lee, 2017), and even though our walls were isolated (i.e., not joined-up) it is similar to the stimuli used in Solstad et al. (2008) where it was demonstrated that border cells are active for walls even when discontinuous with respect to other environment boundaries. Furthermore, our walls impeded movement - a primary feature of boundaries as described previously in the literature (Buckley, Smith, & Haselgrove, 2015; Høydal, Skytøen, Andersson, Moser, & Moser, 2019; Lee, 2017; Lever et al., 2009; Solstad et al., 2008) - and participants spent the majority of their exploration time near the boundaries, which would have made it abundantly clear that they were impassable during learning in the fully-immersive virtual reality setup. That these walls would still be considered as such even in a virtual world is supported by work showing that navigational behaviour during obstacle-avoidance is highly correlated between virtual and real-world setups (Fink, Foo, & Warren, 2007). Distinguishing further our walls from objects, our boundaries did not change position throughout the entire experiment, meaning that that they are experientially different to a transient object. In human behavioural experiments, boundaries influence object-location memory relative to single objects (Negen, Sandri, Lee, & Nardini, 2018), and in fMRI extended boundaries, but not single objects, are associated with hippocampal activity (e.g., Doeller, King, & Burgess, 2008), with this boundary-related medial temporal lobe activity evident when having to imagine even a single boundary (Bird, Capponi, King, Doeller, & Burgess, 2010). Finally, in rodents there is no evidence that subicular BVCs respond to individual objects, and individual objects fail to control the position of place cell firing unless they are arranged to form a boundary (Cressant, Muller, & Poucet, 1997).

Given that there is no published evidence of object responses in BVCs, we are left in a situation where we would have to argue for different neural effects underpinning the same pattern of data in two different ROIs (i.e., above-chance decoding in posterior EC and posterior subiculum). In future studies it will be important to put these two different properties (object versus boundary) in more controlled opposition, but in the current manuscript we have now acknowledged the possible contribution of object vector cells to our decoding accuracy observed in the posterior EC.

Discussion (Lines: 615-631): "A recent study discovered object-vector cells in the rodent MEC that show an allocentric directional response to objects within the

environment⁵¹. Given that both boundary and object-related responses are evident in MEC, and that object-vector cells respond also to boundary-like structures (i.e., elongated objects), it is not possible to distinguish whether our effects are driven by border and/or object-vector cells. Consistent with previous definitions of boundaries, in the current study participants had experience that the walls impeded movement^{10,18,51}, and they comprised an extended 3D surface⁵². Moreover, to our knowledge, there are no reports of object-related firing in the rodent dorsal subiculum. While it is possible that object-vector cells contribute to the posterior EC effect observed in the current study, a more parsimonious explanation is that the walls were considered boundaries and that the decoding performance reflects border and boundary vector cell responses in the EC and subiculum, respectively. The distinction between object-vector and boundary responses in the EC awaits further clarification in humans."

3. Given the lack of significant differences across regions, the main finding is that anterior vs. posterior EC distinction. However, the theoretical interpretation here is not very clear. The introduction (Lines 41–47) reviews the what/where division of LEC/MEC, and then mentions that it has been modified to be about the spatial vs. external sensory inputs (e.g., landmarks and prominent objects). However, the paper that is cited (Knierim et al) does not make that claim, actually; the contrast that is made in that paper is the difference between global and local. If that is the case, it is not very much in line with the current findings, in which the distal landmarks serve as orientation cues and the boundaries could, by some, be considered landmarks or objects. How could this be reconciled or clarified?

We thank the Reviewer for raising this concern and acknowledge that the way in which we described Knierim's work on the MEC versus LEC distinction was misleading. There is considerable evidence now to support a division of labour within the rodent EC, based upon the type of information-to-be represented (e.g., objects/scenes, what/where) or the reference frame in which the spatial information is encoded (allocentric/egocentric). Our analysis in which we segment the EC into posterior and anterior portions, therefore, is justified in its attempt to gain greater insight into this division of labour in humans. We have now reformulated the Background section of the manuscript to emphasise the motivation to segment our ROIs into posterior and anterior portions.

4. As the authors themselves say, the “directional” signals here can be differentiated from the representation of the landmarks themselves (or in the case of the boundaries, the association between the boundaries and the distal landmarks, perhaps). This is quite challenging for the interpretation about goal directions. The lack of varying/counterbalancing of landmark directions (arrangement) should be explained or justified or avoid the interpretation that this is not a directional representation but an object or landmark-related signal.

The Reviewer is correct that while the allocentric boundary direction samples different distal landmark cues, the allocentric goal condition is very much synonymous with this information, meaning that we cannot rule out this interpretation in the current study. The reason that we maintain the same configuration of the landmarks is so that participants continue to have a coherent understanding of the layout of the environment. Given the sparsity of the environment, if we were to change the configuration of landmarks during the experiment, participants would most likely become confused, meaning that we would lose trials through incorrect responses and subsequently it would be more difficult to code trials according to allocentric direction. We have now clarified this in the Discussion.

Discussion (Lines: 670-676): "It was not possible to disentangle the allocentric goal/landmark effect in the current paradigm because we maintained the same configuration of landmarks so that participants had a coherent understanding of the layout of the environment. Given the sparsity of the VE, changing the configuration of the landmarks during the experiment would most likely have confused participants resulting in incorrect responses and therefore lost trials".

5. There should be control analyses to test for egocentric direction (especially given recent reports of such signals in the brain). Similarly, it would be important to show either a comparison between egocentric and allocentric effects. In the boundary condition, the analysis of allocentric boundary directions should exclude the trials in which the allocentric and egocentric direction of the boundary is the same (e.g., when the wall is to the north but you are also facing north at the same time).

We thank the Reviewer for raising these potential confounds, and we have now amended the manuscript to address these points. We have now included the analysis of egocentric boundary and egocentric goal condition. As can be seen in

Supplementary Figure 11, only the anterior subiculum appears to contain information regarding egocentric goal direction; in the remaining EC/subiculum ROIs it is not possible to decode egocentric properties, suggesting that this property is not driving the allocentric effects reported in the manuscript. Furthermore, as can be seen in the trial schematic in Supplementary Figure 2, when decoding either according to allocentric goal or boundary, the egocentric information is sampled equally across different trials. This means that, after the trial averaging over runs, there is no pure egocentric information remaining for these conditions.

6. The results show highly significant differences among the directions (e.g., north vs. east) and an effect of the relative difference between two views (e.g., the difficulty with 270 degrees etc). What is the explanation for these differences? Is there some inherent bias in the environment? Is there any brain activity that differentiates these and explains the neural correlates of the behavioral differences?

The Reviewer is correct that there are differences in RT for the allocentric goal direction task according to direction. Previous studies using judgment of relative direction tasks have demonstrated that participants tend to impose a reference frame on environments, and that this often comprises a conceptual North (Mou & McNamara, 2002). In our study, judgments regarding the allocentric goal direction relative to these axes are faster, and these data could suggest that participants imposed a North-South axis on the environment reflecting the Mountain - Cathedral (as can be seen in Supplementary Figure 7). Importantly, these differences in RT did not explain the decoding performance (see our response to Reviewer 1, point 1), and during learning participants did not show a preference for a particular landmark identity. Why the participants showed poorer performance during learning for the 270 degree angle disparity is unclear. This could in part reflect exploration such that participants explored the environment in a clockwise fashion, and that the information was recapitulated in this way, explaining the increased latency for 270 degree judgments.

We have now also added an interpretation of these effects:

Results (Lines: 497-501): "These differences in RT may reflect participants forming a reference frame in the environment, with the Mountain and Cathedral providing a conceptual North-South axis. Consequently, responses to allocentric goal judgments in these directions may be facilitated."

7. A true boundary-cell-like representation would not distinguish between different walls (the yellow brick wall to the north from the red brick wall to the north). Therefore, it is important to compare between the different “North” boundaries to see if their representations are very similar.

The Reviewer is correct in that border and boundary vector cells are insensitive to the identity of the boundary and therefore neural responses should generalise across the two different boundaries for each cardinal direction. To classify accurately, the linear SVM used here was required to extrapolate over boundary identity, otherwise it would not be possible to distinguish between North/South or East/West allocentric boundary directions due to them sharing the same boundary. Furthermore, the classifier used a one-versus-all approach, meaning that to classify a given direction, it would have to distinguish it from the opposing direction of the same boundary, and generalise over the directional information across the two different walls. Finally, given the drop-out and noise in anterior temporal lobe regions in fMRI, we averaged our trials over runs to boost the signal-to-noise ratio, meaning that we do not have individual responses per boundary side, but instead an averaged response that boosts the condition of interest (e.g., allocentric boundary direction), and reduces the influence of lower-level visual differences (e.g., boundary texture).

8. In humans, in particular, there is quite a bit of lateralization reported in MTL function. Are there any differences when comparing left vs right hemispheres?

We agree that lateralization in medial temporal lobe function has been reported in numerous studies (e.g., Bellmund, Deuker, Schroeder, & Doeller, 2016). To address this concern, we reran the decoding analysis with the medial temporal lobe masks separated by hemisphere and entered the resulting values into an ANOVA comprising the factors Hemisphere, ROI, Anterior/posterior section and Condition. Although there was a significant ROI \times Anterior/posterior interaction ($F(1, 27) = 4.73, p = 0.039$), there was no evidence of any effect of, or interaction with Hemisphere ($ps > 0.81$).

9. The “imagined” part of this task (which is the phase at which the fMRI signal is extracted) seems like an important aspect. Perhaps this should be highlighted from the start.

We thank the Reviewer for this comment. Unlike previous experiments in which participants have been actively instructed to imagine a trajectory (Horner, Bisby, Zotow, Bush, & Burgess, 2016) or the direction between two landmarks (Bellmund et al., 2016), we did not instruct our participants to imagine the boundaries during the presentation of the blank screen. The reason for taking the period of the trial in which there was no stimulus input was to match perceptual features across different classes of stimuli, and not to demonstrate evidence of a more abstract boundary representation. Consequently, we do not want to place too much emphasis on this "imagined" component, but we highlight clearly in the Methods that we use the portion of the trial in which there was no visual input.

Other comments:

10. The authors mention that this is the first demonstration of allocentric boundary direction representation. It seems important to clearly define what they mean by this and how this particular task design aims to distinguish this. Some readers may not understand and, even if they do, they may not follow the logic of how this is not distinguished in other studies).

We thank the Reviewer for this comment, and have now added to the 'Background' section of the manuscript, a clearer definition as to what we mean by us providing the first demonstration of allocentric boundary direction representations.

Background (Lines: 88-92): "Specifically, it does not provide a signal regarding whether the boundary is located to the North, South, East, or West, regardless of the person's position and orientation in the environment."

Background Lines 127-133: "Moreover, unlike previous studies that have demonstrated evidence of increased neural activity in the medial temporal lobe associated with boundary processing, we used multivariate analysis methods to determine whether across multiple voxels there was a neural signature for different allocentric boundary directions (i.e., North, South, East, West)."

11. The literature reviewed is a bit selective and limited. There is now a growing body of evidence that ties together visual scene representations of boundaries (e.g., looking at OPA, PPA) with boundary-based navigation behavior (e.g., reorientation in humans, esp. children, and nonhuman animals).

We thank the Reviewer for suggesting the addition of this literature. We have now amended the Background to include these reorientation effects. Furthermore, we have added a paragraph in the Discussion in which we talk about the relationship between the representations of boundaries in OPA/PPA and our effects in the EC and subiculum.

Background (Lines: 58-59): "The geometric structure of the environment also provides strong cues for reorientation¹³."

Discussion, Lines (695-720): "Although we did not find evidence that the PHC is involved in the processing of allocentric boundary direction, this region has been shown to be exquisitely sensitive to scene stimuli, and in particular the structure of a scene^{60,61}. Scene-selective portions of the PHC discriminate scene stimuli depending upon whether they contain highly visible boundaries regardless of scene content⁶², and add to a network of brain regions including V1 and the lateral occipital cortex that are sensitive to boundary information in scenes⁶³. Furthermore, the occipital place area has been shown to be causally involved in memory for object locations relative to boundaries but not landmarks²¹. It remains to be explored in humans how this lower-level visual information regarding scene structure is combined with the allocentric representations necessary to support allocentric boundary direction coding observed here in the EC and subiculum. One possibility is that representations of these boundary features in the environment are combined, via conjunctive neurons, with both head direction information and egocentric positional estimates relative to the environment walls, as has been demonstrated in rodents⁶⁴. Scene-specific responses have been reported also in the human anterior subiculum⁶⁵. Although these data may seem at odds with our posterior subiculum boundary effects, it is possible that anterior subiculum shows a univariate scene response, whereas the multivariate pattern in posterior subiculum is informative of allocentric boundary information in the absence of greater scene-related activity. Future studies will be necessary to elucidate the nature of scene-sensitivity in the subiculum, and the precise perceptual features driving these effects."

12. Control regions: CA1, CA23DG, and PHG were designated as the control regions, but there is no ANOVA that includes these along with the EC and Sub to ensure that the effects are significantly different in the EC/Sub. Even in the separate analyses in the supplementary section, there were also effects in the

PHG and CA1, but this effect was reported to not survive Bonferroni correction and also not divided along the anterior and posterior axis.

We thank the Reviewer for this comment. Our anatomical ROIs were informed by the rodent literature, where we had clear predictions as to the involvement of the EC and subiculum in boundary coding, particularly with regard to the posterior extent of both of these regions. Consequently, we focussed our analyses in these medial temporal areas. In response to concerns from Reviewer 3 (point 2), we have removed the ROI \times Anterior/posterior \times Condition ANOVA (formerly Figure 5), and present our analysis by ROI only (e.g., Figure 4 - EC and subiculum). Regarding the PHG and CA1, additional analyses in response to Reviewer 3 (point 3) where we erode the masks to reduce the influence of neighbouring brain regions led to the effects in these ROIs no longer being significant; in contrast the effects in EC and subiculum remained relatively unchanged. We acknowledge that our effects are relatively small, and it is likely that other brain regions contribute to this boundary processing. Our statistical tests, however, were used to assess whether the information in a given ROI exceeded that expected by chance, not that decoding performance was significantly larger than would be expected in a different brain region. Consistent with example of grid cell-like representations in humans not being limited to the EC, we would not argue that EC and subiculum are the only regions in which one might observe boundary coding.

13. Line 331-335: the averaging over three runs. Explain any effects across runs.

We thank the Reviewer for allowing us the opportunity to clarify this point. The data were averaged over the three runs to increase signal-to-noise ratio and therefore we did not analyse data per run; the analysis comprised the 96 averaged samples (i.e., 276 trials / 3 runs).

14. Is the boundary usually located on the opposite of the goal direction (if so, doesn't that introduce some bias)? When does the movement end? Where are the goal locations and where are the goals? It would be helpful to have several videos and/or figures showing the paths and the positions of the subject and cue object at test.

We thank the Reviewer for allowing us to clarify the design of the study. The boundary and goal side were counterbalanced such that on half of the trials the goal was located

to the side congruent with the boundary and on the other half of the trials it was on the opposite side. We have now included a schematic figure in the supplementary methods (Supplementary Figure 2) and provide a link to videos in which examples from both the exploration and fMRI scan test phases can be viewed (<https://tinyurl.com/y3otgq7w>).

Reviewer #3 (Remarks to the Author):

Major claims and Novelty

The experimental design, involving prior free exploration and direction judgement task to a strong performance criterion, and subsequent passive viewing that disentangles direction-to-boundary and direction-to-goal, is a strength of the paper. Although the authors don't emphasise it, the design further dissociates location and direction-to-boundary. Thus in the classic circular-walled environments used in many human VR studies, there will only be one location where there is, say, a boundary exactly 2 metres to the East of the subject.

The findings are indeed novel. The main precedents in humans for the current findings are those that the authors already refer to in page 4 of their manuscript. (I refer to the original discoveries of boundary cells in the animal literature below in discussing ROIs.). Environmental Boundary related changes in intracranially recorded theta power have recently been seen in two 2018 studies, when subjects occupy positions (Chen et al, Current Biology, 2018), or consider the location of goals (Lee et al, J Neurosci, 2018), near walls in a virtual environment. Actually, the boundary-related finding in the Chen study was not that theta power increased near walls; rather it was that grid-related hexadirectional modulation of theta power increased near walls. (The present study is also indirectly related to older fMRI evidence showing hippocampal BOLD signal increases associated with imagining walls, and with learning goals defined in relation to boundaries, rather than landmarks). The BVC computational models that predated and predicted physiological BVCs posited that a given BVC fired when a boundary was perceived at a preferred distance and allocentric direction from the subject (e.g. 'when there is a boundary 50cm to the East'). While there remains some debate about the variability of the preferred distance tunings, it is clear that both types of well-characterised boundary cells, i.e. the subicular BVCs and entorhinal border cells, share the property of having a preferred allocentric direction. There is no study to my knowledge reporting a 'direction-

to-boundary' signal in humans other than the Shine et al study under consideration here. They do not study distance to boundary, but this is understandable in an initial study, and the Mosers, for instance, would say that nearly all the boundary cells in the entorhinal cortex fire directly adjacent to the boundary. This is all to say that the current findings are novel, and reveal a signal that is arguably a more direct human analogue of the boundary cells found in rodent exploration than previous findings. This, in my view, is the paper's main claim to originality and significance. As mentioned above, the design allows analysis of separable direction-to-boundary and direction-to-goal signals, which offers an advance over the interesting study of Chadwick et al, (2015, Current Biology). They show some dissociation of the goal and boundary signals, with anterior regions supporting direction-to-goal representations, and this is also novel.

Taken together, subject to various reassurances regarding technical matters, outlined below, I think the novelty and significance of the findings would strongly merit publication in Nature Communications.

Overall fMRI methodology

1. Overall, the fMRI methodology contains apparently essential, beneficial, and perhaps arbitrary choices. I am not familiar enough with fMRI methodology to understand the borderline between these three categories, and what are useful, indeed potentially excellent, innovations, but I would strongly encourage the authors to offer more detail and justification for them, and to appeal to a more general audience. Overall the analysis is quite 'bespoke'. The current supplementary info is rather short, and could be extended as necessary to provide a published rationale for some of their methods. Can the supplementary information be used to illustrate how the main findings survive under different analytical regimes?

We thank the Reviewer for this comment, and would like to reassure them that all of the preprocessing has been used previously in the literature. The basic preprocessing applied to the imaging data are standard steps in fMRI analysis and included in almost all analysis packages. For the multivariate decoding analysis, nearly all analysis steps have been implemented previously in the literature, with the only exception being the bootstrap method to generate the p-value for the group-level decoding accuracy (see response to Reviewer 1, comment number 6, and our clarification of this method in

response to your comment number 4). We have now included Supplementary Figure 5 to clarify our different analysis steps, and provide supporting references in the Methods Section (see response to your comment 5).

Regions of Interest & Lines of analysis

2. As I understand it, the present study is a direct follow up to the rodent literature's well-established findings of boundary cells in the subiculum (boundary vector cells) and in the entorhinal cortex (border cells). These findings have all been replicated by labs not involved in the original reports. Thus, even with the less studied subiculum, there are at least four reports of subicular BVCs including Stewart et al (2014, Trans Phil Roy Soc B); Olson et al (2017, Nature Neuroscience); Brotons-Mas et al (2017, Neuroscience) as well as the data in refs 16 & 18. These papers, and equivalents for the entorhinal border cells, should be cited to document justification for selecting the entorhinal cortex and subiculum as the two ROIs for boundary signals in human fMRI. In summary, there is ample justification for the choice of the two ROIs in the analysis.

In contrast, what is less clear, and needing further a priori rationale, is the posterior/anterior comparison analysis highlighted in the main Figure 5. Given the anatomically basic nature of the division into posterior and anterior portions (i.e. just split at the middle of long axis, see p. 15, lines 309-311), it is not really clear what this abstracted 'posterior' vs 'anterior' result means: there is actually no boundary-vs-goal difference within either of the ROIs, considered singly, and the subiculum and entorhinal cortex are not structurally similar. What precisely were the a priori predictions justifying uncorrected t-tests (lines 387-389), and how many were there? If this cannot be fully justified, perhaps the authors might consider removing this analysis.

We thank the Reviewer for raising this concern and, as suggested, have removed this analysis. The Reviewer is correct in that the anterior/posterior split that we have used is more basic than that used previously with 7T data (e.g., Maass, Berron, Libby, Ranganath, & Düzel, 2015), but this stems in part from the difficulty in implementing this segmentation at lower resolutions. As noted below in response to your comment number 3, although our in-plane resolution was the same as has been implemented previously at higher field strengths, our slice thickness was larger. This means that the transitional slices between anterior-lateral and posterior-medial EC are more likely to

contain a mixture of anatomical information from these subregions. Additional control analyses in which we eroded the masks (see response to your point number 3) to limit the influence of neighbouring ROIs, however, suggest that the results are robust, and the effects we present in the manuscript are not simply an artifact of the segmentation procedure employed.

3. How well can one truly partition the EC, subiculum and other regions CA1, CA23DG, and PHC at 3T with 1.5mm x 1.5mm x 1.5mm voxels? The protocol they mention in reference 32 was developed for 7T imaging (p15, line 308). Even if the anatomical segmentation is accurate in the higher resolution T2 images, it will lose resolution when translated back to the EPI space. For decoding analyses, a very small portion of the sampled data can become heavily weighted if predictive, with large portions more or less ignored. Therefore, the authors should perhaps consider the possibility that above-chance decoding performance can result from 'leakage' from a strongly-involved area (e.g. posterior EC, posterior SUB in boundary direction) to an area that may not actually be that involved (e.g. CA1, anterior EC). In the case of the non-ROIs CA3/2/DG and CA1, this may not matter too much, and these results are not overly emphasised by the authors, with figures in Supplementary information, and regions apart from PHC not surviving multiple corrections. But I think it does bear somewhat upon the PHC, and the anterior EC. How can the authors address these concerns? One is to acknowledge caveats regarding the localisation of the effects. Another, more interestingly, is to offer some anatomical data/figures regarding the locations of highly-weighted voxels. The machine-learning approach should not preclude visual display of the neuroanatomy of activation patterns. Clearly, it is one thing if the highly-weighted PHC voxels are directly adjacent to EC ones, and if highly-weighted anterior EC voxels are clustered near the middle of the entorhinal long axis, and another if highly-weighted voxels are more evenly distributed.

The Reviewer raises an important point regarding the specificity of our ROIs. We would first like to note that although the segmentation protocol used in the current manuscript was established for 7T MRI, the only difference between the resolution used for segmentation in our 3T study lies in the slice thickness; we used a slice 1.5mm thickness whereas the cited protocol used 1mm-thick slices. Importantly, for identifying anatomical boundaries within individual hippocampal subregions, the in-plane resolution is the same in both the protocol and our study (i.e., 0.4*0.4mm).

The Reviewer is correct that there is always a cost in anatomical precision whenever moving between the resolutions used for structural versus functional imaging, and this may lead to partial-volume effects, with voxels containing a mixture of signals from different anatomical regions. Although this situation is, at the current resolutions available, unavoidable in functional imaging, we tried to mitigate the 'leakage' of signal by performing the analysis on unsmoothed data, which we have also now clarified in the Methods (lines 328-330). Although we appreciate the Reviewer's point that weight-maps may provide more anatomical information and help resolve the issue of leakage, currently it is suggested that the presentation of weight maps is not optimal. It could be the case that a voxel with a large weight reflects the removal of a noise signal in the data allowing for the extraction of smaller, but more meaningful signal (Haufe et al., 2014). As a result, highly significant voxels may not actually reflect the neural computations of interest. Also, given the typically limited number of trials available in neuroscience research, many different brain maps will give rise to similar predictive outcomes (Varoquaux & Thirion, 2014). Furthermore, the choice of weight map to be visualized is difficult. As we use nested cross-validation, it is not entirely clear the correct approach to generate a summary weight map, given that each fold of the cross-validation will result in a slightly different model. It is for these reasons that we have chosen not to report the weight maps. Rather, to address the question of leakage more directly, we have re-analysed the data by eroding the masks to reduce the influence of neighbouring anatomical structures.

Eroding our masks removed the outer layer of voxels thereby reducing possible overlap with adjacent ROIs (see Supplementary Figure 4). For our key regions of interest the effects were largely consistent with the original analysis. The only effect that differed from the original manuscript was that allocentric goal decoding in the anterior subiculum was no longer significant ($p = 0.46$; Supplementary Figure 8). Outside of the EC and subiculum, however, it was no longer possible to decode either spatial property in the CA1 or PHC; in CA23DG it was possible to decode allocentric boundary direction ($p = 0.03$; Supplementary Figure 10) but this did not survive Bonferroni correction ($p = 0.008$). This analysis suggests that while the effects reported in our EC/subiculum analysis were unlikely to result from leakage from neighbouring structures, other ROIs may have contributed to the decoding accuracies observed in our additional medial temporal regions. We have now included in the Supplementary Information this additional analysis.

Supplementary Figure 4. *Example of posterior EC and subiculum ROI mask erosion in one participant rendered on mean EPI image.*

Supplementary Figure 8. Mean decoding performance for EC and subiculum ROIs using eroded masks. Our main findings remain relatively consistent even when using these more conservative masks, suggesting that the effects reported in these regions do not reflect leakage of information between adjacent ROIs.

Supplementary Figure 10. Mean decoding performance for medial temporal ROIs using eroded masks. In contrast to the EC and subiculum, the effects in parahippocampal cortex were no longer significant when using more conservative masks.

4. Implementation of bootstrap procedure (p17)

This may reflect my ignorance but it was not fully clear exactly what is meant in this context by group-level decoding accuracy (line 366), group mean decoding accuracy (368) or why this is subtracted from each participant's score before adding chance performance (369). These values are then resampled with another bootstrap to obtain the null distribution, but it's not clear whether this makes sense because of the preceding points. It is not quite clear which distribution is plotted in the pale blue histogram in Figure 4. Another, and perhaps more obvious/conventional, way to determine a null distribution might

be to repeatedly randomize the labels associated with individual trials and then carry out decoding with the shuffled data. Have the authors chosen their approach because it avoids repeatedly retraining the classifier (time-consuming)? More detail and clarity could be provided on these points.

We thank the Reviewer for providing us with the opportunity to clarify the methods used to determine the significance level in our manuscript. In short, we carry out two different bootstrap procedures. The first is to demonstrate the distribution of our sample, the second is to determine the p-value.

The group-level decoding accuracy reflects the mean average decoding across all of our participants. As the Reviewer rightly points out, this is referred to as both "group-level decoding accuracy" and "group mean decoding accuracy". For consistency, in the manuscript we now just refer to this value as "group-level decoding accuracy". This is a single value, which is represented by the vertical black line in Figure 4. The pale blue histogram in these figures reflects the distribution of 10,000 means resulting from bootstrap resampling from the group's individual decoding accuracies. This was computed to demonstrate more clearly the distribution of the sample, rather than simply providing a single value representing the group-level decoding accuracy. The second bootstrap procedure was a Monte Carlo significance test used to determine the p-value associated with the group-level decoding accuracy, and to do this we first needed to generate a null distribution centred around chance performance. Accordingly, we subtracted the group-level decoding accuracy from every individual participant's decoding score (i.e., demeaning the sample) before adding to each participant's demeaned score chance performance (i.e., 25% when decoding four classes). This resulted in the group's decoding scores maintaining the same variance, but with a mean centred on chance. We then sampled from the null distribution 10,000 times and observed how many times the group-level decoding accuracy drawn from this null distribution exceeded the observed mean decoding (i.e., the vertical black line), and divided this number by the number of bootstrap permutations (i.e., 10,000) to obtain our p-value. Importantly, 1 is added to both the numerator and denominator of this calculation to correct for cases in which none of the null values exceed the mean decoding accuracy. We have now clarified this information in the Methods (Lines: 403-424).

Visual after-effects and Control regions

5. The authors will be aware of controversies in the fMRI literature whereby it has been suggested that ostensibly spatial and contextual signals could reflect

uncontrolled visual or path cues (e.g. Nolan et al, 2018, *eneuro*). The issue here is the possibility that analyzed data is affected by visual input during the preceding part of the trial. If this were the case, it might be possible to decode the spatial parameters from the visual information. If I have understood it correctly they avoid this by: i) asking participants to maintain information relevant to the subsequent behavioural decision during a blank screen. It is data from this period that is analysed; ii) a delay of 8 seconds is allowed for hemodynamic lag. I think this is reasonable, although it might be argued that 8s from the onset of the blank screen is not enough for visual activity to die away completely. Can further justification/evidence be provided? It appears they used the general linear model to carry out the regression to remove movement parameters, but then switched to volume averaging to estimate the signal during each trial, and then averaged these across corresponding trials in three different runs. Might it not be better to incorporate the trial or condition regressors into the GLM? I can see their approach might have some advantages in producing more distinct trials for a decoding analysis but it is not well explained in the text. Assuming their approach is justified, as I suspect it can be, it would be best to include an explanatory figure explaining the analysis pipeline, and the rationale should be explained in greater depth either in the main methods or in supplementary information.

Supplementary Figure 5. *fMRI decoding analysis pipeline.*

We thank the Reviewer for providing us the opportunity to explain in more detail the analysis methods used to generate the betas for the decoding analysis. First, with regards to the decoding of visual information, please see below our response to your point number 6. Second, the Reviewer is correct that there are a number of different models that can be used to generate the betas, which have been outlined previously

(Mumford, Turner, Ashby, & Poldrack, 2012). Some researchers have chosen to use separate GLMs for different trials convolved with the haemodynamic response function (HRF) to generate individual beta images. An alternate, and potentially more parsimonious approach in which no prior information regarding the shape of the HRF is fed into the GLM, is to fit an unconvolved boxcar regressor spanning a number of TRs around 4-6 seconds after the event of interest. This so-called "Add" model performed well in both simulations and with real data with short inter-stimulus intervals, which was attributed to its tolerance to the variability of the HRF (Mumford et al., 2012). For our analysis, we used an "Add" model, and mirrored the analysis methods of Bellmund et al. (2016) in which multivariate analysis methods were used to examine the neural response in the entorhinal cortex during an fMRI spatial navigation task. Bellmund et al. (2016) first regressed out the movement parameters from their data before fitting an "Add" model in the residuals resulting from this regression. Consistent with previous research, in a bid to boost the signal-to-noise ratio for the decoding analysis (Isik, Meyers, Leibo, & Poggio, 2013; Nau, Navarro Schröder, Bellmund, & Doeller, 2018), we then averaged the individual trial-specific betas over the three runs and used these resulting estimates for our decoding analyses with nested cross-validation. We have now provided more details in the Methods section and included Supplementary Figure 5 that outlines more clearly this analysis pipeline.

Data analysis (Lines: 361-372): "Given its high performance in decoding using event-related functional imaging data with short inter-stimulus intervals, the "Add"³⁷ model was implemented here. This model aims to capture the putative peak of the haemodynamic response function occurring 4-6 seconds after the onset of the event of interest. Since we wanted to capture activity associated with the stationary period of the trial, which occupied the period 2-6 seconds after trial onset (see Figure 2C), we took the estimates from an unconvolved boxcar regressor that spanned three TRs occurring 4-6 seconds after the stationary phase^{36,38} (i.e., 6-12 seconds after trial onset), in separate models comprising one regressor representing the trial of interest, and a second regressor modelling all other trials in the scan run"

6. Again justification is one thing, but what might be particularly reassuring is to add an analysis from a visual control region (such as V1), even if, given the relatively small slab of brain that they sampled, they cannot sample the whole region. It would be worrying for the interpretation of the results if visual cortex were able to outperform the MTL regions of interest. Simply put, every single region they have looked at, if we ignore multiple corrections for a moment,

produces either a significant boundary direction or goal direction signal. It would be reassuring to report results for an additional control region or two.

As the Reviewer correctly points out, we have only a small slab of brain from which to choose a control region, and recent evidence suggests that regions such as V1, which were previously thought to have little involvement in navigation, also show spatially-modulated responses in the absence of visual input in the rodent brain (Pakan, Currie, Fischer, & Rochefort, 2018) which even includes positional signals (Saleem, Diamanti, Fournier, Harris, & Carandini, 2018). Furthermore, there is increasing evidence of a dynamic interplay between the medial temporal lobe and V1 as evidenced in recent fMRI work (Hindy, Ng, & Turk-Browne, 2016). It becomes difficult, therefore, to be entirely sure that an ROI should show no response to a given task manipulation.

Critically, however, our control analysis on egocentric boundary direction provides reassurance that we are not simply decoding lower-level visual features in EC and subiculum (please see above the response to Reviewer 1, point 2). Specifically, when explicitly introducing a visual confound by classifying trials according to the position of the boundary in the visual field, we could decode egocentric boundary direction in V1. In contrast, decoding accuracy was at chance for both EC and subiculum.

With regards to a significant boundary direction and/or goal direction signal being evident in every ROI, additional analyses highlighted in Supplementary Figures 8 and 10 demonstrate that although our EC and subiculum effects remain relatively consistent using the eroded masks, it is no longer possible to decode either allocentric property in parahippocampal cortex or CA1. These analyses suggest that the effects observed in key EC and subiculum ROIs are robust and not an artifact of our analysis method.

Minor Points

7. Figure 1. Figure 1A. The authors have described the environment as square, but the City-clock axis looks appreciably longer than the mountain-cathedral axis. A key near Figure 1A might be helpful to avoid confusion, saying something like e.g. “W indicates presence of boundary to west”. Figure 1B will need to be larger. The blue tokens and red sensors risk being invisible after shrinking. Legend for sensors should say that they are red. Legend for 1C should state that the landmarks shown are the City and Cathedral.

Landmarks. In the legend for Figure 1, and in methods text, it will greatly help to reader to keep specifying there are four landmarks. Thus, ‘presented with one

of the four landmarks' in figure 1 legend, and 'presented with a static picture of one of the four landmarks' in lines 187-189.

We thank the Reviewer for these helpful suggestions. In the legend to Figure 1a we have now included a key making clear that N, S, E, and W refer to the boundary directions, and have changed the proportions of the schematic to reflect the square environment. We have increased the size of Figure 1B and adjusted the legend in 1C and the Methods to highlight that there were four global landmarks. Furthermore, we have adjusted the schematic of the environment to make it square.

Figure 1. Updated Figure with square environment (a), and increased size of wall sensor and ball token (b).

Results

8. In text (lines 489-90) and figure, the p value for the entorhinal boundary analysis should be given exactly, not $p = 0.0$.

We thank the Reviewer for this comment. We have now increased the number of decimal places in the reporting of the p-value so that the precise number is reported.

9. The behavioural performance paragraphs on pages 20-21 are hard to read. They would be better as tables or graphs. There should be some interpretation of the results, regarding faster reaction times for certain judgements, e.g. for boundaries and goals to the North.

We thank the Reviewer for this comment and note that the accuracy data are displayed in Figure 3C, and the reaction time data in Supplementary Figure 7. Given that the participants performed the allocentric goal direction task, we have provided a brief interpretation for this effect in the Results section. Specifically, in judgment of relative direction tasks, it has been demonstrated that participants impose a reference frame when encoding positional information, and that this is often aligned with geometric cues of the environment, such as room structure, or initial facing direction (Mou & McNamara, 2002). Participants may have interpreted the mountain as a conceptual North, which may have facilitated reaction times for allocentric goal judgements relative to this North-South axis. Importantly, during learning there was no evidence of a landmark preference (Results p.19-20), and there was no evidence that classifier performance was modulated by these differences in RT (see response to Reviewer 1, comment 1).

Results (Lines: 497-503): "These differences in RT may reflect participants forming a reference frame in the environment, with the Mountain and Cathedral providing a conceptual North-South axis. Consequently, responses to allocentric goal judgments in these directions may be facilitated. Consequently, responses to allocentric goal judgments in these directions may be facilitated^{48,49}. Importantly, however, these differences in RT did not influence subsequent decoding performance (see Supplementary Information)".

Discussion

10. P27 – re discussion of Scenes in posterior vs objects in anterior, re EC but implicitly subiculum from the lines above. The authors should consider the work of Hodgett et al, 2017, J Neurosci, on the subiculum, which does not necessarily sit easily within this scheme.

We have now added a brief discussion of this work. We would like to note that although Hodgetts et al. (2017) report scene-selectivity in anterior subiculum, our results do not argue against a univariate scene-preference in anterior subiculum, rather that the

multivariate signal in this region is not informative regarding allocentric boundary direction.

Discussion, Lines (712-720): "Scene-specific responses have been reported also in the human anterior subiculum⁶⁴. Although these data may seem at odds with our posterior subiculum boundary effects, it is possible that anterior subiculum shows a univariate scene response, whereas the multivariate pattern in posterior subiculum is informative of allocentric boundary information in the absence of greater scene-related activity. Future studies will be necessary to elucidate the nature of scene-sensitivity in the subiculum, and the precise perceptual features driving these effects."

11. P27 – the authors should mention other functions for boundaries than error correction for path integration, e.g. defining where objects are relative to boundaries – see TMS work of Julian and Epstein in Current Biology.

We thank the Reviewer for this suggestion and have now incorporated this work in the Background section.

Background (Lines: 75-76): "In humans, boundaries have been shown to be behaviourally salient, aiding reorientation¹³, and being used to define object locations^{20,21}.

Discussion (Lines: 702-704): "Furthermore, the occipital place area has been shown to be causally involved in memory for object locations relative to boundaries but not landmarks"

References

- Bellmund, J. L. S., Deuker, L., Schroeder, T. N., & Doeller, C. F. (2016). Grid-cell representations in mental simulation. *ELife*, 5(AUGUST).
<https://doi.org/10.7554/eLife.17089>
- Besag, J. (1992). Simple Monte Carlo P-Values. In C. Page & R. LePage (Eds.), *Computing Science and Statistics* (pp. 158–162). New York, NY: Springer New York.
- Bird, C. M., Capponi, C., King, J. A., Doeller, C. F., & Burgess, N. (2010). Establishing the Boundaries: The Hippocampal Contribution to Imagining Scenes. *Journal of Neuroscience*, 30(35), 11688–11695.
<https://doi.org/10.1523/JNEUROSCI.0723-10.2010>
- Buckley, M. G., Smith, A. D., & Haselgrove, M. (2015). Learned predictiveness training modulates biases towards using boundary or landmark cues during navigation. *Quarterly Journal of Experimental Psychology*, 68(6), 1183–1202.
<https://doi.org/10.1080/17470218.2014.977925>
- Chadwick, M. J., Jolly, A. E. J., Amos, D. P., Hassabis, D., & Spiers, H. J. (2015). A Goal Direction Signal in the Human Entorhinal / Subicular Region. *Current Biology*, 25, 1–6. <https://doi.org/10.1016/j.cub.2014.11.001>
- Cressant, A., Muller, R. U., & Poucet, B. (1997). Failure of Centrally Placed Objects to Control the Firing Fields of Hippocampal Place Cells. *Journal of Neuroscience*, 17(7), 2531–2542. <https://doi.org/10.1523/JNEUROSCI.17-07-02531.1997>
- Doeller, C. F., King, J. A., & Burgess, N. (2008). Parallel striatal and hippocampal systems for landmarks and boundaries in spatial memory. *Proceedings of the National Academy of Sciences*, 105(15), 5915–5920.
<https://doi.org/10.1073/pnas.0801489105>
- Fink, P. W., Foo, P. S., & Warren, W. H. (2007). Obstacle Avoidance During Walking in Real and Virtual Environments. *ACM Trans. Appl. Percept.*, 4(1).
<https://doi.org/10.1145/1227134.1227136>
- Haufe, S., Meinecke, F., Görgen, K., Dähne, S., Haynes, J.-D., Blankertz, B., & Bießmann, F. (2014). On the interpretation of weight vectors of linear models in multivariate neuroimaging. *NeuroImage*, 87, 96–110.
<https://doi.org/10.1016/j.neuroimage.2013.10.067>
- Hindy, N. C., Ng, F. Y., & Turk-Browne, N. B. (2016). Linking pattern completion in the hippocampus to predictive coding in visual cortex. *Nature Neuroscience*, 19(5), 665–667. <https://doi.org/10.1038/nn.4284>

- Hodgetts, C. J., Voets, N. L., Thomas, A. G., Clare, S., Lawrence, A. D., & Graham, K. S. (2017). Ultra-High-Field fMRI Reveals a Role for the Subiculum in Scene Perceptual Discrimination. *The Journal of Neuroscience*, *37*(12), 3150–3159. <https://doi.org/10.1523/JNEUROSCI.3225-16.2017>
- Hope, A. C. A. (1968). A simplified Monte Carlo significance test procedure. *Journal of the Royal Statistical Society: Series B (Methodological)*, *30*(3), 582–598.
- Horner, A. J., Bisby, J. A., Zotow, E., Bush, D., & Burgess, N. (2016). Grid-like processing of imagined navigation. *Current Biology*, *26*(6), 842–847. <https://doi.org/10.1016/j.cub.2016.01.042>
- Høydal, Ø. A., Skytøen, E. R., Andersson, S. O., Moser, M.-B., & Moser, E. I. (2019). Object-vector coding in the medial entorhinal cortex. *Nature*, *568*(7752), 400–404. <https://doi.org/10.1038/s41586-019-1077-7>
- Isik, L., Meyers, E. M., Leibo, J. Z., & Poggio, T. (2013). The dynamics of invariant object recognition in the human visual system. *Journal of Neurophysiology*, *111*(1), 91–102. <https://doi.org/10.1152/jn.00394.2013>
- Lee, S. A. (2017). The boundary-based view of spatial cognition: a synthesis. *Current Opinion in Behavioral Sciences*, *16*(August), 58–65. <https://doi.org/10.1016/j.cobeha.2017.03.006>
- Lever, C., Burton, S., Jeewajee, A., O'Keefe, J., & Burgess, N. (2009). Boundary Vector Cells in the Subiculum of the Hippocampal Formation. *Journal of Neuroscience*, *29*(31), 9771–9777. <https://doi.org/10.1523/JNEUROSCI.1319-09.2009>
- Maass, A., Berron, D., Libby, L. A., Ranganath, C., & Düzel, E. (2015). Functional subregions of the human entorhinal cortex. *ELife*, *4*(JUNE), 1–17. <https://doi.org/10.7554/eLife.06426>
- Mou, W., & McNamara, T. P. (2002). Intrinsic Frames of Reference in Spatial Memory. *Journal of Experimental Psychology: Learning Memory and Cognition*, *28*(1), 162–170. <https://doi.org/10.1037/0278-7393.28.1.162>
- Mumford, J. A., Turner, B. O., Ashby, F. G., & Poldrack, R. A. (2012). Deconvolving BOLD activation in event-related designs for multivoxel pattern classification analyses. *NeuroImage*, *59*(3), 2636–2643. <https://doi.org/10.1016/j.neuroimage.2011.08.076>
- Nau, M., Navarro Schröder, T., Bellmund, J. L. S., & Doeller, C. F. (2018). Hexadirectional coding of visual space in human entorhinal cortex. *Nature Neuroscience*, *21*(2), 188–190. <https://doi.org/10.1038/s41593-017-0050-8>
- Negen, J., Sandri, A., Lee, S. A., & Nardini, M. (2018). Boundaries in Spatial Cognition: How They Look is More Important than What They Do. *BioRxiv*.

<https://doi.org/10.1101/391037>

- Pakan, J. M. P., Currie, S. P., Fischer, L., & Rochefort, N. L. (2018). The Impact of Visual Cues, Reward, and Motor Feedback on the Representation of Behaviorally Relevant Spatial Locations in Primary Visual Cortex. *Cell Reports*, 24(10), 2521–2528. <https://doi.org/10.1016/j.celrep.2018.08.010>
- Saleem, A. B., Diamanti, E. M., Fournier, J., Harris, K. D., & Carandini, M. (2018). Visual Cortex and Hippocampus. *Nature*, 562(7725), 124–127. <https://doi.org/10.1038/s41586-018-0516-1>
- Solstad, T., Boccara, C. N., Kropff, E., Moser, M.-B., & Moser, E. I. (2008). Representation of Geometric Borders in the Entorhinal Cortex. *Science*, 322(5909), 1865–1868. <https://doi.org/10.1126/science.1166466>
- Varoquaux, G., & Thirion, B. (2014). How Machine Learning Is Shaping Cognitive Neuroimaging. *Gigascience*, 3, 28. <https://doi.org/10.1186/2047-217x-3-28>